# Widespread stable noncanonical peptides identified by integrated analyses of ribosome profiling and ORF features

Haiwang Yang[1,3], Qianru Li[1,3], Emily K. Stroup [1], Sheng Wang[2] & Zhe Ji [1,2] ✉

Studies have revealed dozens of functional peptides in putative 'noncoding' regions and raised the question of how many proteins are encoded by non-canonical open reading frames (ORFs). Here, we comprehensively annotate genome-wide translated ORFs across five eukaryotes (*human*, *mouse*, *zebra-fish*, *worm*, and *yeast*) by analyzing ribosome profiling data. We develop a logistic regression model named PepScore based on ORF features (expected length, encoded domain, and conservation) to calculate the probability that the encoded peptide is stable in *humans*. Systematic ectopic expression validates PepScore and shows that stable complex-associating microproteins can be encoded in 5′/3′ untranslated regions and overlapping coding regions of mRNAs besides annotated noncoding RNAs. Stable noncanonical proteins follow conventional rules and localize to different subcellular compartments. Inhibition of proteasomal/lysosomal degradation pathways can stabilize some peptides especially those with moderate PepScores, but cannot rescue the expression of short ones with low PepScores suggesting they are directly degraded by cellular proteases. The majority of *human* noncanonical peptides with high PepScores show longer lengths but low conservation across species/ mammals, and hundreds contain trait-associated genetic variants. Our study presents a statistical framework to identify stable noncanonical peptides in the genome and provides a valuable resource for functional characterization of noncanonical translation during development and disease.

Conventional genome annotation has largely been based on computational analyses of gene structure and sequence conservation using the "one-gene one-polypeptide" hypothesis[1,2]. The canonical protein-coding sequence of an mRNA has been defined as the longest open reading frame (ORF) with an AUG start codon and is under evolutionary constraint. Alternative splicing and poly-adenylation can generate mRNA isoforms with different coding regions. Genes without a long and/or conserved ORF have usually been classified as long noncoding RNAs (lncRNAs) or pseudogenes based on the assumption that long proteins are more likely to fold into stable structures with biological functions[3–6]. One conventional approach to defining coding regions is using an arbitrary cutoff such as 100 aa just because ORFs longer than this occur rarely in a genome[7]. Alternatively, computational methods were developed to predict the "coding potential" of a transcript[8,9], but these models were trained on molecular features of canonical proteins with a median length of 500 aa showing species conservation. Although these methods can identify functional proteins, they ignore ORFs encoding stably folded microproteins or species-specific peptides[10].

[1]Department of Pharmacology, Feinberg School of Medicine, Northwestern University, Chicago, IL 60611, USA. [2]Department of Biomedical Engineering, McCormick School of Engineering, Northwestern University, Evanston, IL 60628, USA. [3]These authors contributed equally: Haiwang Yang, Qianru Li. ✉e-mail: zhe.ji@northwestern.edu

Noncanonical ORFs (ncORFs) have historically been overlooked, in part due to the technical challenge of defining actively translated ones. If we only consider ORF structures, there are millions of possible ORFs in a eukaryotic genome. Recent advances in computational analyses of ribosome profiling[11,12] have resolved this technical barrier. For actively translated ORFs, ribosome profiling reads show continuous 3-nucleotide (nt) periodicity, representing the stepwise movement of 80 S ribosomes as they decode RNAs. Based on this read distribution feature, we and others have developed computational algorithms to identify genome-wide translated ORFs using ribosome profiling[13–17]. These analyses revealed thousands of translated ncORFs in a cell encoded by annotated lncRNAs, pseudogenes, 5' untranslated regions (UTRs), and 3'UTRs.

The comprehensive identification of translated ncORFs paved the path to systematically dissecting their biological roles. CRISPR screening experiments showed that knocking out some ncORFs inhibited cell proliferation[18,19]. Ectopic expression of these ncORFs showed that hundreds of noncanonical peptides, including poorly conserved ones, can be stably expressed in cells[18–20]. Mass spectrometry experiments showed thousands of these peptides are displayed by the major histocompatibility complex I (MHC I) and can function as neoantigens for disease therapy, although many of them are possibly unstable and quickly degraded in cells[21–23].

Importantly, emerging genetic studies have revealed detailed molecular functions of dozens of microproteins. For example, the secreted hormonal microproteins ELABELA (APELA, 55 aa) is conserved across vertebrates and regulates embryonic stem cell self-renewal[24] and heart development[25]. A few microproteins, such as phospholamban (PLN, 55 aa), sarcolipin (SLN, 31 aa), and myoregulin (MRLN, 46 aa), regulate sarcoplasmic reticulum $Ca^{2+}$ ATPase activity and muscle function[26–28]. A putative *human* lncRNA LINC00992 encodes a GATA3-interacting cryptic protein (GT3-INCP, 120 aa) that promotes estrogen receptor-positive breast cancer progression[29].

As high stability is crucial for protein functions, the findings demand the development of computational methods to distinguish ncORFs encoding stable proteins vs. quickly degraded byproducts. Here, we aimed to address this question by analyzing large cohorts of ribosome profiling data and comprehensively annotating genome-wide translated ORFs across five eukaryotes from *yeast* to *humans*. Furthermore, based on the genomic features of ORFs encoding characterized microproteins, we developed and experimentally validated a machine learning model named PepScore to calculate the probability that an ORF-encoded peptide is stable, which provides a statistical framework to study noncanonical peptides.

## Results

### A comprehensive catalog of translated ORFs across five eukaryotic species

Large cohorts of published ribosome profiling datasets across model species provided a valuable resource to examine the expression and evolution of ncORFs. A challenge for the data analyses is that sequencing reads from different studies show quite variable distribution patterns due to RNase digestion differences or lineage-specific regulation. Only a subset of reads showed 3-nt periodicity in translated regions and can be used to accurately identify actively translated ORFs (Supplementary Fig. 1a, b). To this end, we further developed our RibORF software[13,30] (RibORF 2.0), which automates the performance of data quality control, selects reads showing 3-nt periodicity, and uses ribosomal A-site corrected reads to identify genome-wide ORFs (Fig. 1a).

In total, we analyzed 38.8 billion reads across 669 *human* samples, 559 *mouse* samples, 34 *zebrafish* samples, 43 *worm* samples, and 7 *yeast* samples (Supplementary Data 1 and Fig. 1b). We selected 3.1 billion reads showing strong 3-nt periodicity in canonical ORFs (>60% assigned to the first nt of codons) (Fig. 1c, Supplementary Fig. 1c–f, and

Supplementary Data 1). Our RibORF trained a logistic regression model to identify translated ORFs showing continuous 3-nt periodicity across codons based on the following read distribution features: (1) the fraction of reads assigned to the 1st nt of codons; (2) the fraction of codons showing 3-nt periodicity and supporting in-frame translation; and (3) the percentage of maximum entropy (PME) value measuring the uniformness of reads across codons (Supplementary Fig. 2a–c; see Methods for details).

Our logistic regression model can accurately classify in-frame vs. off-frame ORFs with an area under the receiver operating characteristic (AUROC) curve of 0.990 (SD = 0.011) across samples (Supplementary Fig. 2d and Supplementary Data 2), and the three modeling features we used contributed significantly to the prediction (Fig. 1d–f and Supplementary Fig. 2e–h). To account for lineage-specific regulation, we built a predictive model for each cell type from a ribosome profiling dataset as well as a separate model using the merged reads from a species (Supplementary Data 2). We required that retained ORFs should show at least two independent positive predictions with the RibORF translation probability >0.6, and the ORF isoforms should express in at least one cell type with reads per kilobase per million mapped reads (RPKM) > 0.2 calculated by the Salmon tool[31].

The analyses of the compendium of ribosome profiling datasets provided comprehensive annotations of the translatome (Fig. 1g, Supplementary Data 3, and Supplementary Data 4). In *humans*, we identified 58,383 ncORFs in 13,062 coding genes, 3887 annotated lncRNA genes, and 1287 pseudogenes. We cross-validated our ORF predictions using other published software, including RiboTish[32], RiboCode[33], PRICE[34], and RiboTricer[35]. If requiring the perfect match of predicted ORF structures, 98.3% of our ncORFs ≥20 aa and 88.0% of these <20 aa can be identified by at least one other software (Supplementary Fig. 3a and Supplementary Data 5). Because these software use different methods to select the representative start codon of an ORF[16], if we allow start codon mismatch of predictions, the validation rate for our ncORFs is 99.8% for ORF ≥ 20 aa and 97.0% for ORFs <20 aa. This high cross-validation rate is consistent for different ncORF types (Supplementary Fig. 3b–h), indicating the high-quality of our ORF annotation.

73.5% of *human* coding genes showed translation outside canonical ORFs (Fig. 1h). 62.8% were translated in 5'UTRs with 28,981 upstream ORFs (uORFs) and 4679 overlapping uORFs (ouORFs), and 19.6% showed translation in 3'UTRs with 4907 downstream ORFs (dORFs) and 738 overlapping dORFs (odORFs) (Fig. 1h and Supplementary Fig. 4a–c). 56.7% of ncORFs used AUG as start codons (Supplementary Fig. 4b, c). A comparable fraction of coding genes in *mouse* showed noncanonical translation (51,481 ncORFs identified in total; Fig. 1h and Supplementary Fig. 4bc). We detected noncanonical translation in 37% of *zebrafish* coding genes and 15% of genes in *worm* and *yeast* (Fig. 1h). The number of identified ncORFs in mammals is 2.5-fold that in *zebrafish*, 20-fold that in *worm*, and 50-fold that in *yeast* (Supplementary Fig. 4b, c). One reason is that *human* and *mouse* have more high-quality ribosome profiling datasets available, and the other reason is that mammals have more annotated lncRNAs and their coding genes have longer 5'/3' UTRs (1.2- to 1.8-fold) with more transcript isoforms per gene (2.2- to 2.7- fold) (Supplementary Fig. 4d). These results underscored the complexity of mammalian translatome.

### Identifying genomics features of stable microproteins

The translation of ncORFs can produce stable peptides or quickly degraded byproducts. Currently, there lack of computational approaches that can distinguish these two types of ncORFs. Addressing this question is important because high stability is crucial for protein function. To this end, we examined the genomic features of ORFs encoding experimentally characterized microproteins (<100 aa) with an AUG start codon. We collected 343 stably expressed peptides from two sources: (1) RefSeq-curated microproteins, which were

shown to be stably expressed and have well-annotated molecular functions, such as metallothionein 1X (MT1X, 61 aa) and NADH:ubiquinone oxidoreductase subunit A1 (NDUFA1, 70 aa); and (2) stably detectable peptides encoded by annotated lncRNAs from recent ectopic expression experiments (Fig. 2a and Supplementary Fig. 5a; Supplementary Data 6; see Methods for details)[18,19]. A previous study also identified 100 microproteins undetectable from ectopic expression (Supplementary Data 6)[18], and these peptides are possibly unstable as their ORF expression were driven by the same 5'UTR and 3'UTR compared to the detectable ones. We used them as negative examples for comparison. We focused on AUG-initiated ORFs in this analysis, because 5'-end methionine is known to be critical for stabilizing proteins[36] and the vast majority of known stable peptides use AUG as the start codon.

Existing studies have prioritized conserved proteins for functional characterization, and the collection of stable microproteins showed higher conservation levels measured by PhyloCSF scores (based on genome alignments across 58 mammals)[37] than undetectable ones (Fig. 2b). We next divided stable peptides into three groups based on their PhyloCSF scores: >0, −5 ~ 0, and <−5. The peptides from all three groups showed comparable lengths (median: 78 aa) and were

significantly longer than undetectable ones (median: 51 aa) (Fig. 2c; Wilcoxon Rank Sum Test $P < 10^{-9}$). These results suggested that ORF length is an important parameter that identifies stable peptides, in addition to conservation.

We hypothesized that long ORFs would not emerge by chance in a genome and that randomly occurring ORF structures would be short. We aimed to calculate the false discovery rates (FDRs) of observed ORF lengths by comparing them to expected ones from randomized sequences. Considering RefSeq-defined coding genes, protein lengths are positively correlated with transcript lengths ($R = 0.71$; Supplementary Fig. 5b), and transcripts encoding microproteins are significantly shorter than those producing long proteins (Wilcox rank sum test $P = 2 \times 10^{-12}$; Supplementary Fig. 5c). To obtain the expected ORF length of a transcript, we controlled the transcript length and generated pseudo-transcripts from shuffled sequences and random genome sequences. Then we calculated the FDR of an observed ORF length as the ratio of the number of ORF structures longer than the observed vs. the total number of possible ORFs from pseudo-transcripts (Supplementary Fig. 5d).

Indeed, randomly occurring ORF structures are short (Supplementary Fig. 5e) and the expected ORF lengths are correlated with

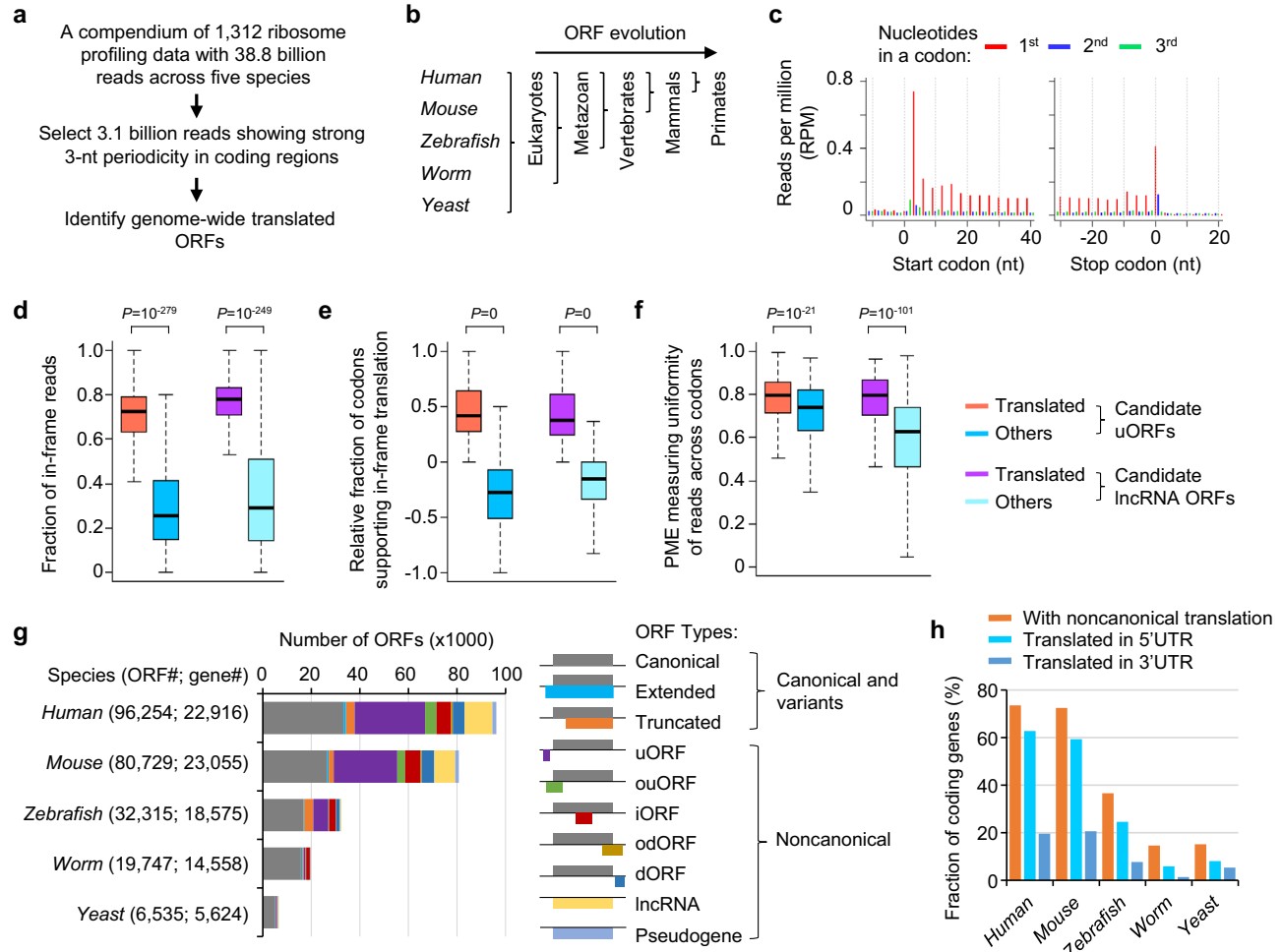

**Fig. 1 | Genome-wide translated ORFs identified in five eukaryotic species by analyzing ribosome profiling datasets. a** Steps of ribosome profiling data analyses. **b** The five eukaryotic species included in the analyses and their phylogenetic relationship. **c** The ribosomal A-site adjusted reads used to identify genome-wide translated ORFs in *humans*. **d–f** The read distribution features used to distinguish translated ORFs vs. other candidate ORFs: the fraction of in-frame reads (**d**), the relative fraction of codons supporting in-frame translation (**e**), and PME measuring uniformity of read distribution (**f**). The boxes are bounded by the 25 and 75 percentiles and the center represents the median. The whiskers extend from each edge of the box to indicate the 1.5x interquartile range. We randomly sampled 1000 ORFs in each group for comparison. The two-sided Wilcoxon Rank Sum Test *P*-values comparing the translated ORFs vs. other candidate ORFs are shown. **g** The statistics of translated ORFs identified across species, grouped by transcript type and ORF location. **h** The fraction of protein-coding genes with noncanonical translation.

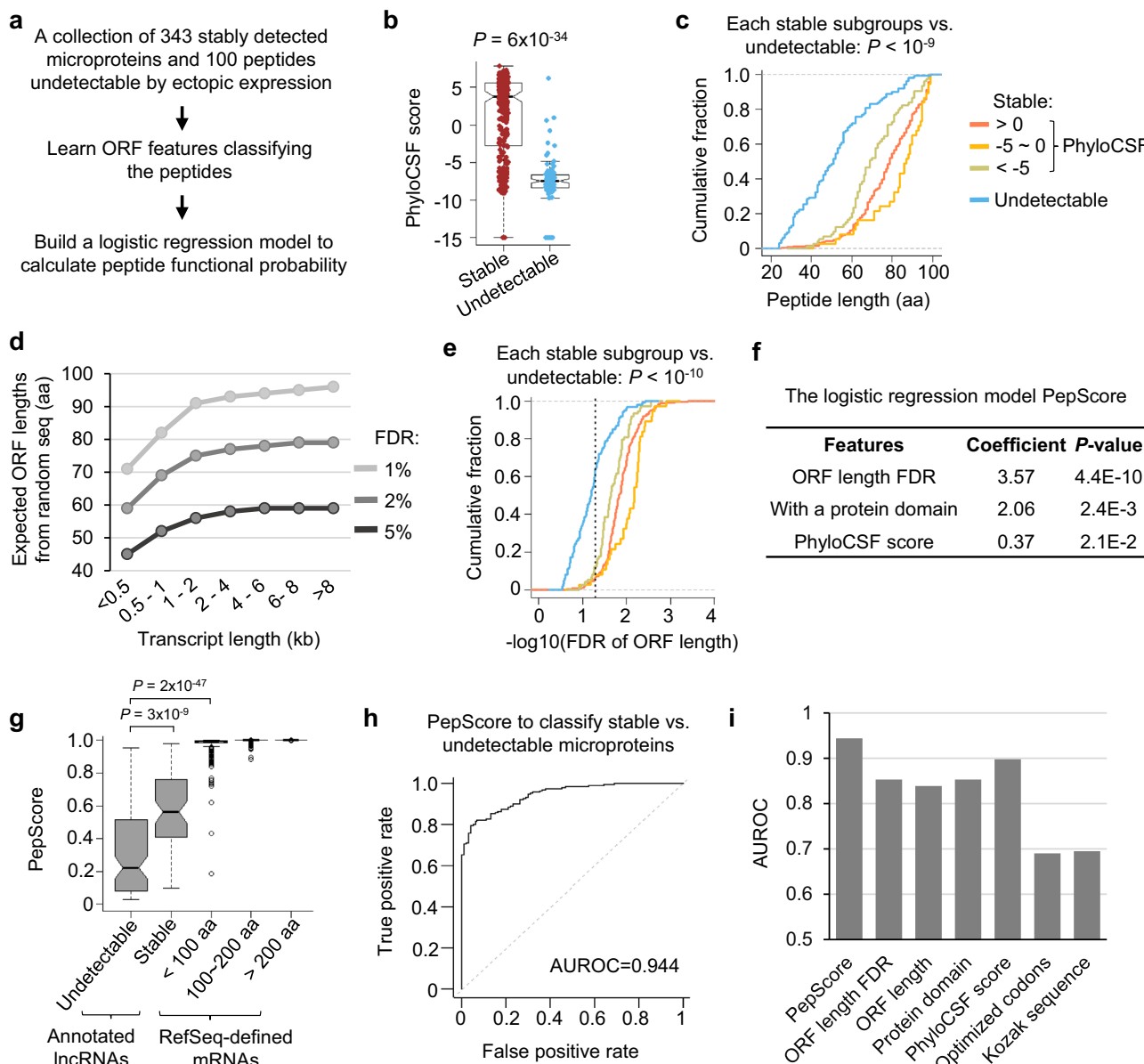

**Fig. 2 | A logistic regression model, PepScore, predicts the stable probability of noncanonical peptides. a** Overview of data analysis steps. **b** Distribution of PhyloCSF scores of stable ($N = 343$) vs. undetectable ($N = 100$) microproteins. The two-sided Wilcoxon Rank Sum Test $P$-value is shown. **c** Cumulative lengths of stable vs. undetectable microproteins. The stable ones were divided into three groups based on their PhyloCSF scores and were compared with the undetectable peptides: $< -5$ ($N = 73$, $P = 2 \times 10^{-10}$); $\geq -5$ and $\leq 0$ ($N = 37$, $P = 3 \times 10^{-12}$); $>0$ ($N = 233$, $P = 6 \times 10^{-24}$). The $P$-values were calculated using the two-sided Wilcoxon Rank Sum Test. **d** The expected ORF lengths at different FDRs based on randomized transcript and genome sequences. We grouped transcripts based on different length ranges for the calculation. **e** The FDRs of observed ORF lengths in stable vs. undetectable microproteins. As in (**c**), the stable ones were divided into three groups based on their PhyloCSF scores and were compared with the undetectable peptides: $< -5$ ($N = 73$, $P = 3 \times 10^{-11}$); $\geq -5$ and $\leq 0$ ($N = 37$, $P = 7 \times 10^{-13}$); $>0$ ($N = 233$, $P = 1 \times 10^{-25}$). The

$P$-values were calculated using the two-sided Wilcoxon Rank Sum Test. **f** The logistic regression model PepScore classifies stable vs. undetectable microproteins. The coefficients and $P$-values of the training parameters are shown. **g** The PepScore distribution of indicated peptide groups. The boxes are bounded by the 25 and 75 percentiles and the center represents the median. The whiskers extend from each edge of the box to indicate the 1.5x interquartile range. $N = 99$ for undetectable microproteins and $N = 67$ for stable ones from annotated lncRNAs; $N = 273$ for RefSeq-defined proteins <100 aa, $N = 4318$ for proteins between 100 aa and 200 aa, and $N = 27{,}566$ for protein >200 aa. The $P$-values calculated using the two-sided Wilcoxon Rank Sum Test are shown. **h** The ROC curve showing the PepScore performance to classify stable vs. undetectable microproteins. The AUROC value is shown. **i** AUROC values for various models using different parameters to classify stable vs. undetectable microproteins.

transcript length (Fig. 2d and Supplementary Fig. 5f). The median transcript length for the collection of 343 stable microproteins was 1 kb. The expected ORF length at a 5% FDR was 55 aa and that at 2% FDR was 73 aa. For the corresponding observed ORF lengths, 92% showed FDRs <5% and 64% were <2%. For the ORFs encoding stable peptides with different PhyloCSF scores, their length FDRs were comparable and were more significant than the undetectable ones (Fig. 2e). These

analyses indicated that many ORFs encoding stable microproteins are longer than expected when considering their transcript lengths.

We next examined other ORF features (Supplementary Fig. 6). RefSeq-defined microproteins used more optimized codons and stronger Kozak sequences around initiation sites compared to undetectable ones (Supplementary Fig. 6c, d; Wilcoxon Rank Sum test $P = 10^{-8}$). Using the hydrophobic cluster analysis (HCA) method[38,39], we

examined the folding potential of the peptides. RefSeq-defined microproteins showed a moderately higher HCA measured foldability (Supplementary Fig. 6e; Wilcoxon Rank Sum test $P = 10^{-3}$). However, these features did not show significant differences between detectable vs. undetectable microproteins based on the ectopic expression experiments as most of these peptides are encoded by annotated lncRNAs (Supplementary Fig. 6c–e).

Additionally, we searched for protein domains encoded by the characterized peptides, including Pfam[40] and TMHMM[41]. 97.8% of RefSeq-defined microproteins and 15.5% of stably detectable peptides from annotated lncRNAs contained at least one functional domain, while only 6.8% of undetectable ones did (Supplementary Fig. 6m). A domain sequence would not emerge by chance in a peptide.

## A logistic regression model named PepScore to predict peptide stable probability

Our above analyses identified ORF features separating stable vs. undetectable microproteins. Next, we used these features to build a logistic regression model named PepScore to calculate the probability that an ORF encodes a stably expressed peptide. We used the above collection of ORFs encoding 343 stable and 100 undetectable microproteins as positive and negative examples, respectively. We also added 30 ORFs encoding ≤11 aa peptides but with PhyloCSF scores >0 as negative examples. We randomly selected 200 peptides for the training dataset with 5-fold cross-validation, and another 100 peptides for the testing dataset.

We used different feature combinations to classify the stable vs. undetectable microproteins. Three of the ORF features contributed significantly to our final model. The length FDR showed the most significant contribution to the classifier ($P = 4.4 \times 10^{-10}$). The existence of a protein domain contributed the second most ($P = 0.0024$), and the $P$-value for the PhyloCSF score was 0.021 (Fig. 2f). Adding additional ORF features did not increase the prediction accuracy. PepScore accurately classified microproteins with an overall AUROC = 0.944. The classification accuracy based on the feature combination is higher than using individual ones (Fig. 2g–i).

The two positive sets of stable microproteins (i.e. RefSeq-defined and stably detectable from ecotopic expression) all show significantly higher PepScores than the negative set: $P = 10^{-10}$ comparing detectable vs. undetectable from ectopic expression and AUROC = 0.79 classifying the two groups; $P = 10^{-46}$ comparing RefSeq-defined vs. undetectable and AUROC = 0.99 classifying the two groups (Supplementary Fig. 6n–p). 99.3% of RefSeq-defined microproteins (<100 aa) showed a PepScore >0.6 including very short ones such as RPL41 (25 aa), SLN (31 aa), PLN (52 aa), MRLN (46 aa), and APELA (54 aa) (Fig. 2g). Because of the logistic regression algorithm we used, PepScore can be applied to calculate the stable probability for long proteins. The RefSeq-defined proteins (>100 aa) show scores close to 1 (Fig. 2g).

## Most noncanonical peptides with high PepScores tend to be poorly conserved

Next, we calculated PepScores for RibORF-identified translated ORFs (with AUG start codons) in *humans* (Supplementary Fig. 7a–f; Supplementary Data 7). 4812 ncORFs showed PepScores >0.6, which is 13.3% of the total (Fig. 3a). These included 1871 ORFs in annotated lncRNAs, 987 ORFs in pseudogenes, 687 uORFs, 430 ouORFs, 188 internal ORFs (iORFs), 239 odORFs, and 410 dORFs (Fig. 3b and Supplementary Data 7). The median length of these peptides was 99 aa with the 5th percentile at 57 aa and the 95th percentile at 278 aa (Fig. 3c). 80.9% of noncanonical peptides showed a low PepScore (<0.3), suggesting that most noncanonical translation generates quickly degraded byproducts.

85.6% of high-PepScore (>0.6) peptides showed a PhyloCSF score <0, indicating their low conservation across mammals (Fig. 3d and Supplementary Data 7). 34.0% contained at least 1 predicted protein

domain (Fig. 3e). Considering the conservation of both ORF types and expression, 18.0% of these high PepScore peptides were conserved in *mouse*, <1.5% were conserved in *zebrafish* and *worm*, and none was conserved in *yeast* (Fig. 3f, g and Supplementary Data 7). The data indicated that most noncanonical peptides are uniquely expressed in *humans* or *mammals*. 3011 of these ncORFs showed differential expression >10-fold across 11 tissue/cell lineages (Supplementary Fig. 7g). Compared to canonical ones, ncORFs showed higher tissue-specific expression as measured by Tau-index values[42] (Wilcox rank sum test $P = 10^{-82}$; Fig. 3h and Supplementary Fig. 7g, h). Using DeepLoc[43], we predicted the subcellular localization of these peptides. Compared to canonical proteins, noncanonical ones showed a 3.4-fold enrichment of localization to mitochondria (36.9% of the total) and extracellular secretion (13.7% of the total) (Fig. 3i).

We next searched the noncanonical peptides by analyzing published cohorts of mass spectrometry data for the whole proteome and for MHC I-bound peptides. Using a cutoff of FDR < 1%, we detected 326 noncanonical peptides from the whole proteome and 1480 bound by the MHC. The peptides expressed from the whole proteome showed significantly higher PepScores and were longer than MHC-bound ones (Wilcox rank sum test $P = 10^{-82}$; Fig. 3j and Supplementary Fig. 7i). We also analyzed the published CRISPR/Cas9 screening data[19] measuring cell survival after knocking out the translated regions in lncRNAs and uORFs in induced pluripotent stem cells (iPSCs) and K562 cells. The loss of ncORFs with high PepScores (>0.6) resulted in stronger cell growth inhibition compared peptides with low PepScores (≤0.6) (Fig. 3k and Supplementary Fig. 7jk), although the two groups showed comparable PhyloCSF scores (Supplementary Fig. 7l). These data indicate ncORFs with high PepScores are more likely to be stably expressed and play functional roles.

## Systematic ectopic expression validated the correlation between PepScore and peptide stability

Since PepScore was trained based on the features of annotated main ORFs from mRNAs as well as those from lncRNAs, we next examined its accuracy for identifying stable proteins from polycistronic mRNAs. To this end, we performed ectopic expression of 29 randomly selected uORF peptides with different PepScores in the HEK293T cells (Fig. 4a, b, Supplementary Fig. 8a, b, and Supplementary Data 8). To compare their relative stability, we fused each ORF with the same 5′/3′ UTR sequences and a C-terminal Flag tag (8 aa)[44], and used immunostaining experiments to examine the expression of the peptides across single cells. As the translation of these ORFs were driven by the same 5′/3′ UTRs with an AUG start codon, the resulting peptide expression differences would more reflect the regulation of peptide stability, but not the ORF translatability. Using green fluorescent protein (GFP) expression to control cell transfection efficiency, we calculated the peptide expression level as the fraction of cells showing detectable expression (Fig. 4c).

Peptides with high PepScores showed stronger expression than low score ones ($P = 0.02$, Wilcoxon Rank Sum test comparing those with PepScore >0.6 vs. others) (Fig. 4d, e). For the nine peptides with PepScore >0.6, four (44%) expressed in >50% of cells, three (33%) expressed between 5–12%, and the remaining ones (22%) were detectable in ~1.5% of cells (Fig. 4d and Supplementary Data 9). For the 12 peptides with PepScores between 0.3 and 0.6, only one (8%) showed expression in >5% of cells, four (33%) expressed between 1% and 4%, and the remaining ones (58%) were nearly undetectable (Fig. 4d). We could not detect expression for the eight peptides with PepScore <0.3 (Fig. 4d). PepScore can classify well-expressed peptides (>5% of cells) vs. others with an AUROC = 0.911 (Fig. 4f) and with a classification accuracy that is higher than using individual features such as ORF length or PhyloCSF score (ΔAUROC > 0.071; Fig. 4g).

We next examined whether the inhibition of the conventional protein degradation pathway can stabilize uORF peptides. To this end,

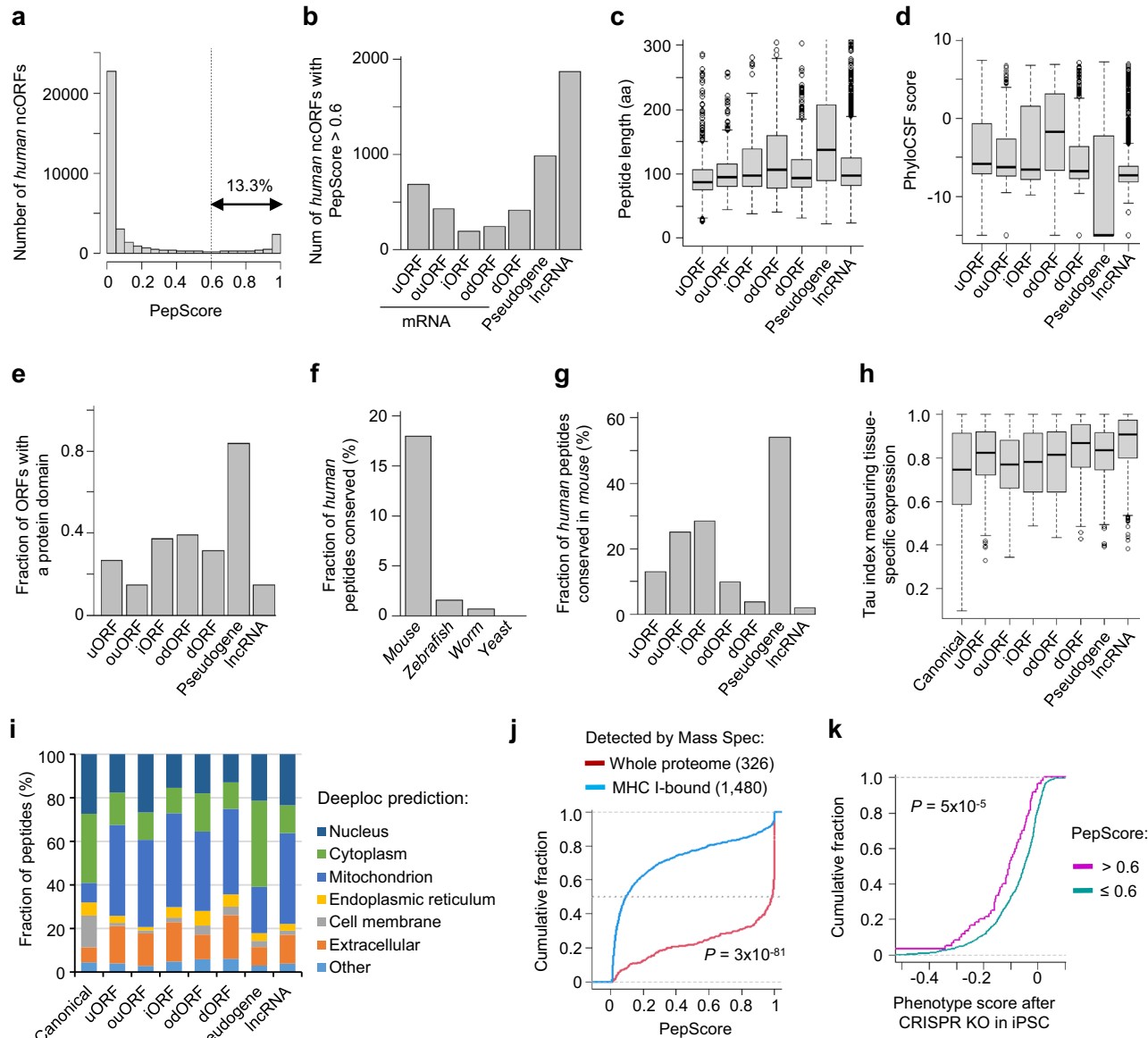

**Fig. 3 | The features of ncORFs with high PepScores. a** The PepScore distribution of *human* ncORFs. **b** Number of ncORF types with PepScore > 0.6. These high-PepScore ORFs were used in the following panels (**c**–**i**). **c** The length distribution of ncORFs. The boxes are bounded by the 25 and 75 percentiles and the center represents the median. The whiskers extend from each edge of the box to indicate the 1.5x interquartile range. The numbers of ncORFs used in the plot are shown in (**b**). **d** The PhyloCSF score of ncORFs. The boxplot format is the same as described in (**c**), and the numbers of ncORFs are shown in (**b**). **e** Fraction of high-PepScore ncORFs with a protein domain. **f** Fraction of the ncORFs conserved in different species. **g** Fraction of the ncORFs conserved in *mouse*, grouped by ORF types. **h** The Tau index measuring the tissue-specific expression of the indicated ORF types. The boxplot format is the same as described in (**c**). The numbers of ncORFs are shown in (**b**), and 33,238 canonical ORFs were analyzed for the comparison. **i** The DeepLoc-predicted peptide localization, grouped by ORF type. **j** The distribution of Pep-Scores for the peptides detected by mass spectrometry, including 326 peptides detected in the whole proteome and 1480 bound by MHC I. The *P*-value calculated using the two-sided Wilcoxon Rank Sum Test is shown. **k** The distribution of phenotype scores after CRISPR knockout of the ORFs in iPSCs. The ORFs were grouped by PepScore: 60 ncORFs with high PepScores (>0.6) and 854 ncORFs with low PepScores (≤0.6). The *P*-value calculated using the two-sided Wilcoxon Rank Sum Test is shown.

we treated the cells with the proteasome inhibitor MG-132 and found that the peptide response was also correlated with PepScore (Fig. 4d, i, and Supplementary Data 9). For the 15 peptides expressed in <6% of cells and with PepScore >0.3, 11 (73%) showed higher expression upon MG-132 treatment (the average expression increased from 2.1% (SD = 2.1%) to 14.3% (SD = 4.9%)) (Fig. 4d). For the eight peptides with Pep-Score <0.3, we could not detect their expression even after MG-132 treatment (Fig. 4d). These data indicate that noncanonical peptides with moderate PepScores can be stabilized upon proteasome pathway inhibition.

To further examine pathways regulating noncanonical peptide stability, we treated cells with a few other inhibitors, including those blocking proteasome activity (lactacystin) or lysosome/autophagy-mediated pathways (chloroquine and bafilomycin A1) (Fig. 4h). Individual treatment with these drugs stabilized noncanonical peptides with moderate PepScores, and combination treatment further enhanced the peptide expression (Fig. 4j, k and Supplementary Fig. 8c, d). These data indicate that the stability of these peptides is regulated by both the proteasome and lysosome/autophagy pathways. Peptides with low PepScores (<0.3) showed nearly

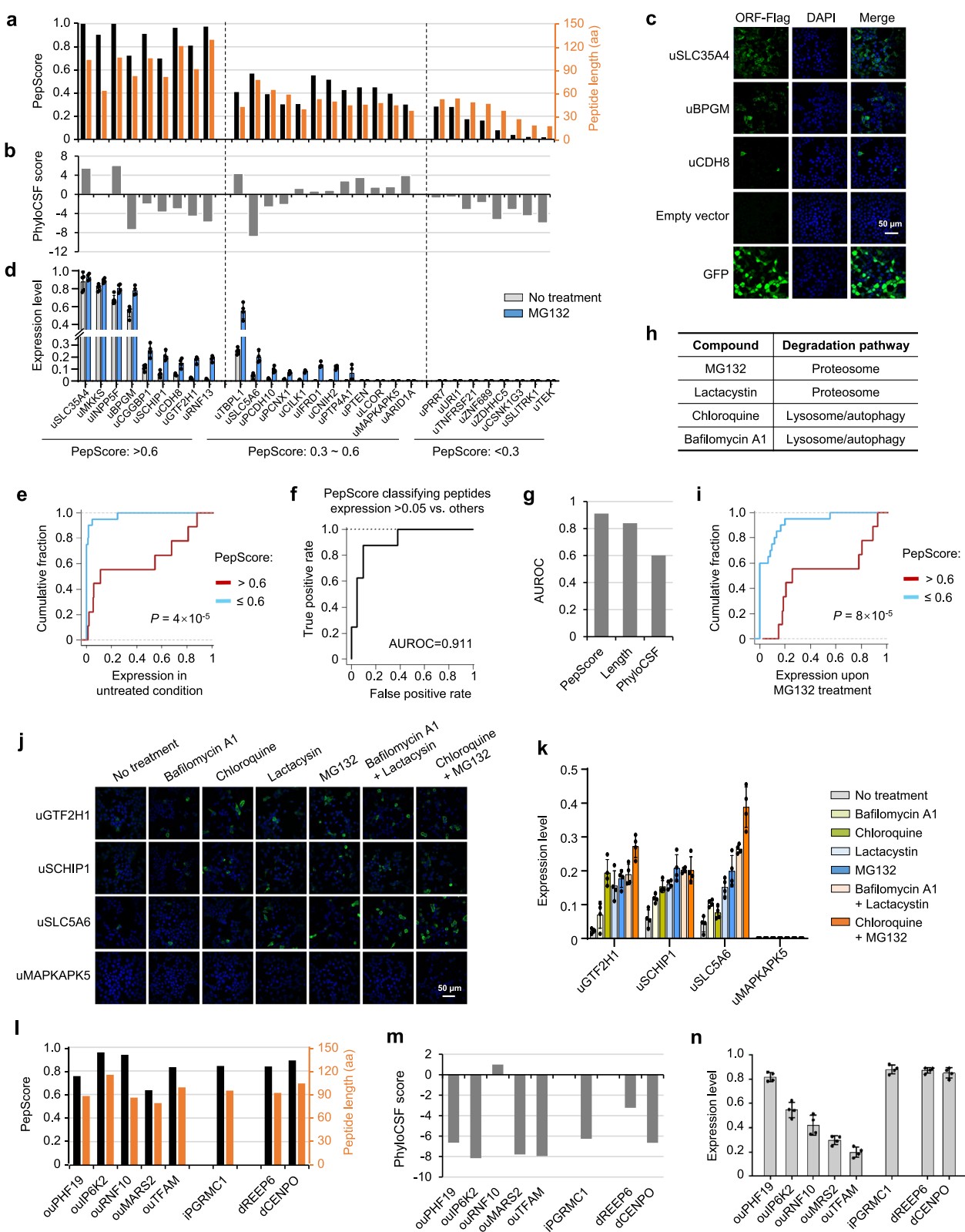

undetectable expression even after the combination treatments (Fig. 4j, k and Supplementary Fig. 8c, d). One possible explanation for this is that these short peptides can be directly digested by proteases without the facilitation of the proteasome and lysosome/autophagy pathways.

We also examined the expression of other ncORF types and performed ectopic expression of the eight peptides with high PepScores (>0.6) from overlapping coding regions and 3′UTRs, including five ouORFs, one iORF, and two dORFs (Fig. 4l, m and Supplementary Fig. 8e–g). These peptides were well expressed in >20% of cells (Fig. 4n). Altogether, our data indicate that stable proteins can be encoded across different regions of mRNAs and that PepScore provides a quantitative framework to identify candidate stable noncanonical peptides.

**Fig. 4 | The stability and degradation pathways of ncORF peptides with different PepScores. a** PepScores and peptide lengths of 29 selected uORFs. **b** The PhyloCSF scores of 29 selected uORFs. **c, d** The ectopic expression of uORFs in HEK293T cells. Cells with ORF-Flag expression were stained with anti-Flag (green) and DAPI (blue). Empty vector and GFP-Flag were used as the negative and positive controls, respectively. Representative images of cells expressing selected uORFs are shown in (**c**). Scale bar, 50 μm. The *y*-axis represents the normalized fraction of cells expressing ORF-Flag. Data are shown as mean values ± SD of five (nontreatment) or four (MG132 treatment) replicates and are representative of three independent experiments. **e** Comparing peptide expression of uORFs with high vs. low PepScores in untreated cells. The *P*-value calculated using the two-sided Wilcoxon Rank Sum Test is shown. **f** The ROC curve measuring the performance using PepScore to classify uORFs with expression >5% vs. others. The AUROC value is shown. **g** The AUROC values obtained when using PepScore, peptide length, and PhyloCSF

to classify highly vs. lowly expressed uORF peptides. **h** Compounds used in this study and their targeted pathways. **i** Comparing peptide expression of uORFs with high vs. low PepScores in MG132-treated cells. The *P*-value calculated using the two-sided Wilcoxon Rank Sum Test is shown. **j–k** The uORF peptide expression levels were analyzed using untreated cells or those treated with proteasome inhibitors or/ and lysosome inhibitors. Representative immunostaining images of selected uORF peptides in each condition are shown in (**j**). Scale bar, 50 μm. The expression levels are shown in (**k**). Error bars represent the standard deviation of four replicates. **l** PepScores and peptide lengths of five selected ouORFs, one iORF, and two dORFs. **m** The PhyloCSF scores of selected ncORFs. **n** The ectopic expression levels of the noncanonical peptides. The calculation method is the same as in (**d**). **k, n** Data are shown as mean values ± SD of four replicates and are representative of three independent experiments. The peptide expression level can be found in Supplementary Data 9.

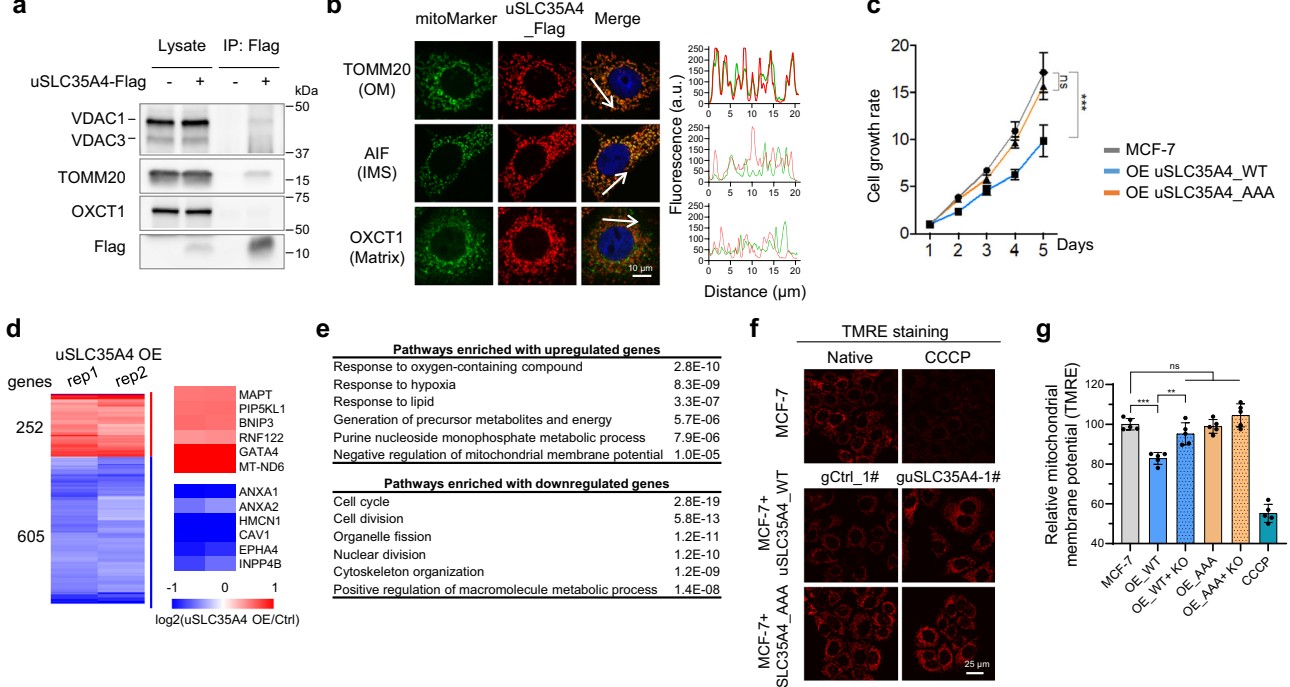

**Fig. 5 | uSCL35A4 is a mitochondrial outer membrane protein and regulates cell proliferation and mitochondrial membrane potential. a, b** uSLC35A4 protein interacts with mitochondrial outer membrane proteins. Immunoblotting analysis of uSLC35A4 co-IP lysates (**a**). Co-immunostaining analysis of uSLC35A4 protein using uSLC35A4_Flag stably expressing MCF-7 cells (**b**). VDAC1/3 and TOMM20 are mitochondrial outer membrane (OM) markers, AIF is the inner membrane space (IMS) marker, and OXCT1 is the matrix marker. Pseudocolored confocal images are shown on the left. Scale bar, 10 μm. The intensity profiles of Flag (red) and indicated mitochondrial marker (green) along the white arrows in the merged images are shown on the right. **c** Overexpression of wild-type but not start codon-mutated uSLC35A4 impairs MCF-7 Cell growth. Data are shown as mean values ± SD of four replicates. (***\**\**P* < 0.001, *P* = 0.0009; ns, not significant, *P* = 0.2452; two-tailed *t*-test). **d** Heatmap showing differentially expressed genes in uSLC35A4_Flag stably expressing MCF-7 cells. The cells expressing start codon-mutated (AAA instead of

AUG) ORF sequences were used as the control. Blue: down-regulated; Red: up-regulated. OE, overexpression. **e** Gene ontology analyses of differentially expressed genes in (**d**). **f, g** The mitochondrial membrane potential of MCF-7 cells measured by TMRE (tetramethylrhodamine, ethyl ester) staining. CCCP (a mitochondrial oxidative phosphorylation uncoupler) was used as system control. Representative confocal images are shown in (**f**). Scale bar, 25 μm. The statistic results are shown as mean values ± SD of five replicates in (**g**). Compared with control, wide-type uSLC35A4 overexpression (OE_WT) decreased MMP (***\**\**P* < 0.001, *P* = 0.0003), which was rescued by uSLC35A4 knock out (KO) (*\**P* < 0.01, *P* = 0.0023). Cell growth was unaffected by overexpressing mutated uSLC35A4 (OE_AAA) or a combination with KO (ns not significant, two-tailed *t*-test). Experiments (**a–c**, and **f–g**) were performed three times with similar results. Source data are provided as a Source Data file.

## uSLC35A4 and iPGRMC1 are mitochondrial proteins with different biological roles

Proteins are localized to specific subcellular compartments to fulfill biological functions. We used the DeepLoc program (v1.0)[43] to predict the subcellular localization of the stable noncanonical peptides examined above. Consistent with the prediction, the immunostaining experiments showed that five peptides, including uSLC35A4, uMKKS, iPGRMC1, uCGGBP1, and outTFAM, are localized to mitochondria (Supplementary Fig. 9a). The endogenous expression and

mitochondrial localization of uMKKS were also shown in a previous study[45]. Although uSLC35A4 was suggested to be a functional protein[46], its molecular roles remain uncharacterized. We confirmed the expression of uSLC35A4 protein from the polycistronic RNAs by expressing the native transcript sequence and tagging both the uSLC35A4 and downstream main ORF (Supplementary Fig. 9b). The mutation of uSLC35A4 start codon diminished the uSLC35A4 expression but increased the SLC35A4 protein production, indicating the uORF suppresses the main ORF translation (Supplementary Fig. 9b).

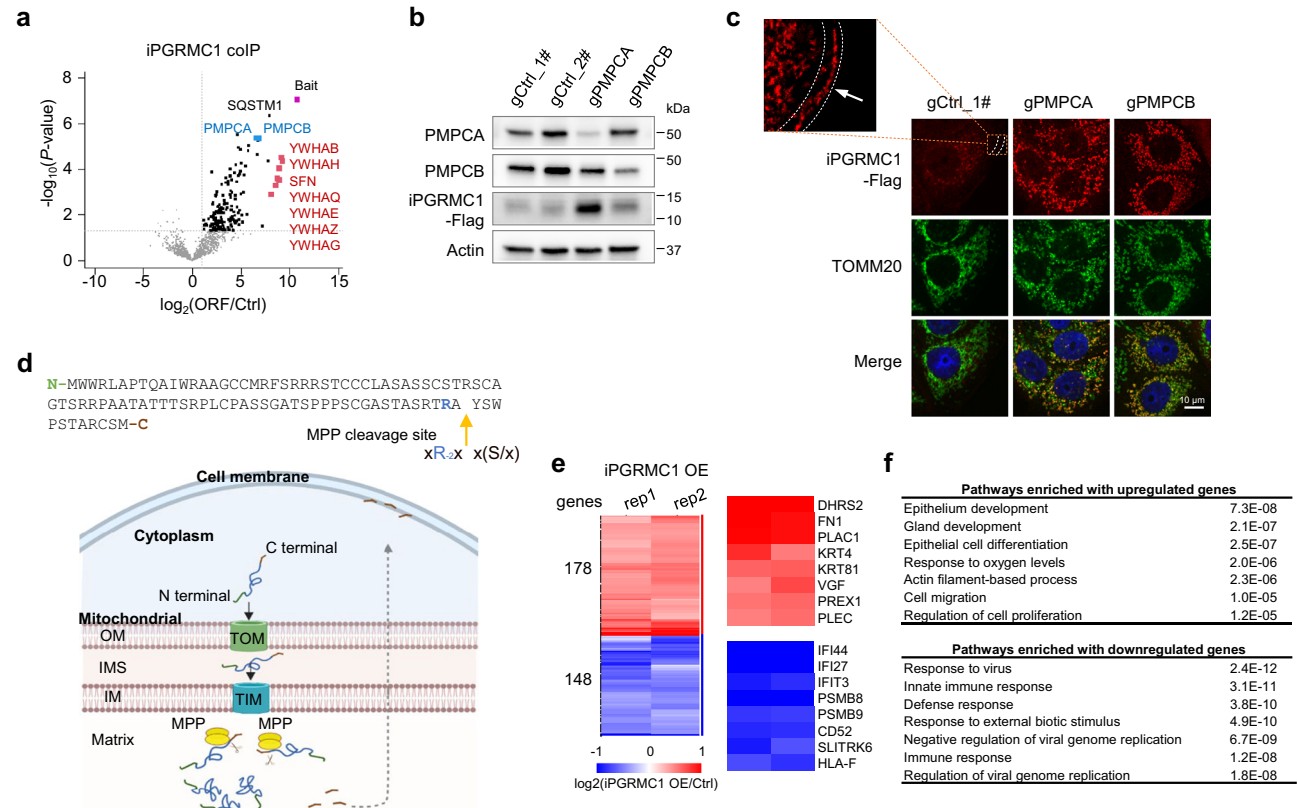

**Fig. 6 | iPGRMC1 is a mitochondrial matrix protein and is cleaved by mitochondrial processing peptidases. a** Volcano plot showing proteins enriched in iPGRMC1 co-IP lysates vs. control whole cell lysates (two-sided *T*-test, *n* = 3 independent experiments). We highlighted the bait and top interacting proteins, including MPP subunits PMPCA and PMPCB (blue), and 14-3-3 proteins (red). **b**, **c** Knocking out PMPCA or PMPCB rescues the detection (**a**) and the mitochondrial localization (**b**) of iPGRMC1. The coimmunostaining of TOMM20 was used to examine the mitochondrial localization. Scale bar, 10 μm. In control cells, the iPGRMC1 peptide shows cell membrane localization, indicated by the orange arrow. Experiments were performed three times with similar results. Source data are provided as a Source Data file. **d** Schematic overview of mitochondrial processing of iPGRMC1 by MPP. The upper panel shows the predicted MPP cleavage site by the R-2 motif. The lower model shows that iPGRMC1 peptides are transported into mitochondria through the translocase of the outer membrane (TOM) and the translocase of the inner membrane (TIM) complexes. After processing by MPP in the matrix, C-terminal iPGRMC1 peptides translocate to the cell membrane. The image was created with BioRender.com. **e** Heatmap showing differentially expressed genes in iPGRMC1 stably expressing MCF-7 cells. The cells expressing start codon-mutated ORF sequences were used as the control. Blue: down-regulated; Red: up-regulated. **f** Gene ontology analyses of differentially expressed genes in (**e**).

Using co-immunoprecipitation mass spectrometry (co-IP/MS) and western blot, we showed that the uSLC35A4 protein interacts with mitochondrial outer membrane proteins, such as VDAC1 and VDAC3 (Fig. 5a and Supplementary Fig. 9c, d). The co-immunostaining experiments showed that uSLC35A4 was colocalized with the outer membrane marker TOMM20 but not other subcompartment markers (Fig. 5b). To further examine its cellular roles, we generated MCF-7 cells stably expressing uSLC35A4 using a lentiviral system, with 3-fold overexpression compared to endogenous levels (Supplementary Fig. 9e–g). Compared to the start codon-mutated sequence, the uSLC35A4 overexpression impaired cell growth (Fig. 5c). In line with this, RNA sequencing (RNA-seq) analyses showed that 605 genes were down-regulated upon uSLC35A4 overexpression (>1.3-fold in both replicates) and were enriched in pathways such as "cell cycle" and "cell division" (Fig. 5d, e). Conversely, 252 genes were up-regulated and were enriched in the mitochondria-related pathways such as "response to hypoxia" and "negative regulation of mitochondrial membrane potential" (Fig. 5d, e). Indeed, overexpression of uSLC35A4 induced the loss of mitochondrial membrane potential, which was rescued by knockout of uSLC35A4 (Fig. 5f, g and Supplementary Fig. 9h). Altogether, we identified that uSLC35A4 is a mitochondrial outer membrane protein, the expression of which regulates cell proliferation and mitochondrial membrane potential.

We next examined the cellular roles of iPGRMC1. We confirmed the expression of iPGRMC1 in the native transcript context by expressing the internal ORF but not disrupting the protein sequence of the main ORF (Supplementary Fig. 9i and Supplementary Data 8). The mutation of the iPGRMC1 start codon didn't affect the expression of canonical PGRMC1 expression (Supplementary Fig. 9i). Compared with cells stably expressing uSLC35A4 or uMKKS, we found cells stably expressing iPGRMC1 showed a much lower peptide expression confirmed by both western blotting and immunostaining (Supplementary Fig. 9e, f). The co-IP/MS identified two major interacting complexes: mitochondrial-processing peptidase (MPP, consisting of PMPCA and PMPCB) and the molecular chaperone 14-3-3 proteins (composed of seven subunits, e.g., YWHAB and YWHAH) (Fig. 6a and Supplementary Fig. 9j).

We hypothesized that the iPGRMC1 peptide might be processed by MPP in mitochondria. We thus set up knockouts of PMPCA and PMPCB in iPGRMC1 stably expressing cells. Indeed, either PMPCA or PMPCB knockout increased the full-length iPGRMC1 peptide level (Fig. 6b) and its mitochondrial localization (Fig. 6c). In addition, we found that iPGRMC1 showed detectable cell membrane enrichment, which was abolished upon PMPCA or PMPCB knockout (Fig. 6c and Supplementary Fig. 9a), suggesting its C-terminal peptide might be translocated to the cell membrane after MPP processing. To further

verify this, we generated a stable cell line expressing dual-tagged iPGRMC1 with an N-terminus HA-tag and a C-terminus Flag Tag (Supplementary Fig. 9k). We detected two smaller peptides (~10 kD and ~2 kD) generated after MPP cleavage with western blotting (Supplementary Fig. 9k). Co-immunostaining of cells with both tags showed that the N-terminus fragment accumulated in the cytosol, while its C-terminus was enriched on the cell membrane (Supplementary Fig. 9l). In line with this, peptide sequence analysis identified an MPP cleavage site motif (the "R-2 rule") at the C-terminus[47,48] (Fig. 6d). Besides, we performed RNA-seq to examine differential gene expression upon stable iPGRMC1 overexpression in MCF-7 cells (2-fold overexpression compared to endogenous levels) (Supplementary Fig. 9e–g). Gene ontology analysis showed that up-regulated genes were enriched in pathways such as "epithelium development" and "cell migration," and down-regulated genes were enriched in pathways related to "innate immune response" (Fig. 6e, f). Taken together, our results indicate that iPGRMC1 is cleaved by mitochondrial-processing peptidase generating an 11-mer peptide translocated to the cell membrane.

### Noncanonical peptides follow conventional sequence determinants and are localized to different organelles

Besides mitochondrial peptides, we examined ORFs encoding peptides localized to other organelles. uINPP5F, uBPGM, and ouMRS2 peptides were localized to the endoplasmic reticulum (ER) (Fig. 7a), and their interaction with ER marker proteins (e.g., calnexin and BIP) was also supported by co-IP followed by western blotting (Fig. 7b). We also confirmed the uINPP5F and uBPGM expression in the native transcript context with main ORF (Fig. 7c and Supplementary Fig. 10a, b). The mutations of the uORF start codons increased main ORF protein production, indicating the uINPP5F and uBPGM suppresses main ORF translation (Fig. 7c).

Based on DeepLoc prediction, uINPP5F is localized to the ER. uBPGM and ouMRS2 peptides were predicted to be secreted extracellularly, but their ER localization is consistent with the fact that most extracellular proteins tend to be processed in the ER before secretion. Furthermore, we isolated exosomes from the cell media and could detect the expression of uBPGM, ouMRS2, and uINPP5F in them (Fig. 7d). Co-IP/MS on these three proteins showed that their top interacting proteins were enriched with these ER-localized or secretory proteins (Supplementary Fig. 10c–e). For example, uINPP5F interacts with secretory vesicle membrane proteins such as ATP6V0A1, ATP6V0D1, and POMGNT1 (Supplementary Fig. 10c).

We also confirmed the cytosolic and/or nuclear localization of uCDH8, ouPHF19, ouIP6K2, ouRNF10, dCENPO, and dREEP6 peptides with immunostaining and/or co-IP MS (Fig. 7e and Supplementary Fig. 10f, g). Of those, we further confirmed the expression and peptide localization of ouPHF19 and dCENPO in their native transcript context, and the start codon mutations of these two ncORFs did not impact the protein production from corresponding main ORFs (Fig. 7f, g and Supplementary Fig. 10h, i). Altogether, our results indicate that ncORF-encoded proteins follow the basic molecular properties of canonical proteins for subcellular localization to carry out diverse cellular roles.

### ncORFs with high PepScores contain disease/trait-associated variants

Genetic studies have revealed sequence variants associated with *human* diseases or traits. Most of these variants are located outside of canonical coding regions and were historically considered as 'noncoding'. Since we identified >50,000 novel translated ORFs across the *human* genome, we next examined whether genetic variants can cause peptide mutations in these ORF regions. To this end, we collected variants annotated by the ClinVar and GWAS databases[49,50], selected those located outside of RefSeq-defined coding regions, and intersected them with the ncORFs identified in this study (Fig. 8a). In total,

6,621 ClinVar and 585 GWAS variants were located in ncORFs (Fig. 8b, c, Supplementary Fig. 11a, b, and Supplementary Data 10). 77% of these variants were nonsynonymous mutations causing amino acid substitution, start/stop codon change, or frameshift (Fig. 8b, c). Among AUG-initiated ORFs with high PepScores (>0.6), 212 were from ClinVar and 142 were from GWAS (Fig. 8d).

For example, the inflammasome subunit NLRP3 gene encodes an AUG-initiated uORF peptide (length = 98 aa and PepScore = 0.977). This uORF contains 11 ClinVar SNPs, including six missense, one stop gained, and four synonymous variants (Fig. 8e). The associated diseases include familial cold autoinflammatory syndrome, chronic infantile neurological cutaneous and articular syndrome, and familial amyloid nephropathy with urticaria and deafness.

We examined the annotated risks for the ClinVar variants in ncORFs. 41.4% were annotated as "uncertain significance", and 49.3% were considered as "benign" or "likely benign" (Supplementary Fig. 11c). These risk annotations were mostly based on their impact on disrupting known functional elements in the genome. Future work is needed to examine the functional roles of these ncORFs, which may reveal novel pathogenic roles of some variants.

## Discussion

Recent advances in ribosome profiling data analysis revealed pervasive translation in putative 'noncoding' regions. Emerging studies have shown that some microproteins, including lowly conserved ones, are stable with biological functions. These findings demand the comprehensive identification of translated ORFs across genomes and the development of computational approaches to distinguish noncanonical translation events generating stable peptides vs. translation byproducts. Here we collected a large cohort of published ribosome profiling datasets and selected high-quality reads showing strong 3-nt periodicity to identify genome-wide translated ORFs across five eukaryote species: *human, mouse, zebrafish, worm,* and *yeast*. Compared to the existing ORF databases[51,52] and studies of ORF evolution[13,53,54], our annotation leveraged more ribosome profiling datasets across species and we required that the ncORFs annotated in this study should show 3-nt periodicity of read distribution.

Our study represents the first to develop a logistic regression model, PepScore, that calculates the probability that the peptide encoded by a noncanonical is stable. PepScore was trained on a collection of experimentally characterized microproteins. Our systematic follow-up experiments validated the correlation between PepScore and peptide stability. Three molecular features contributed significantly to the PepScore: the ORF length FDR, conservation, and the presence of a protein domain in a peptide sequence. Studies examining ORF evolution[13,53,54] have mostly relied on the conservation of ORF structure and peptide sequences to identify candidate functional proteins. As a result, many well-studied microproteins are conserved. Unexpectedly, the peptide length contributed most significantly to the PepScore. Different from using an arbitrary cutoff of 100 aa, we calculated the FDR of each observed ORF sequence. Many microproteins showed significant FDRs when controlling for their transcript length. This is consistent with the fact that long proteins are more likely to be stably folded, and >60 aa seems to be a cutoff based on our analyses (Fig. 2c).

Considering ncORFs with high PepScores, most of them showed significant length FDRs but with low conservation levels. For 4812 *human* ncORFs with high PepScore (>0.6) and AUG start codons, 85.6% showed negative PhyloCSF scores indicating their poor conservation across mammals. 80% of these *human* peptides are not conserved in any of the other four species we examined. And 18% are conserved only between *human* and *mouse*, but not in *zebrafish, worm,* or *yeast*. Importantly, many of these lowly conserved peptides can be stably expressed in the cells. For 16 noncanonical peptides showing an expression level >5% in our ectopic expression experiments, 12 (75%)

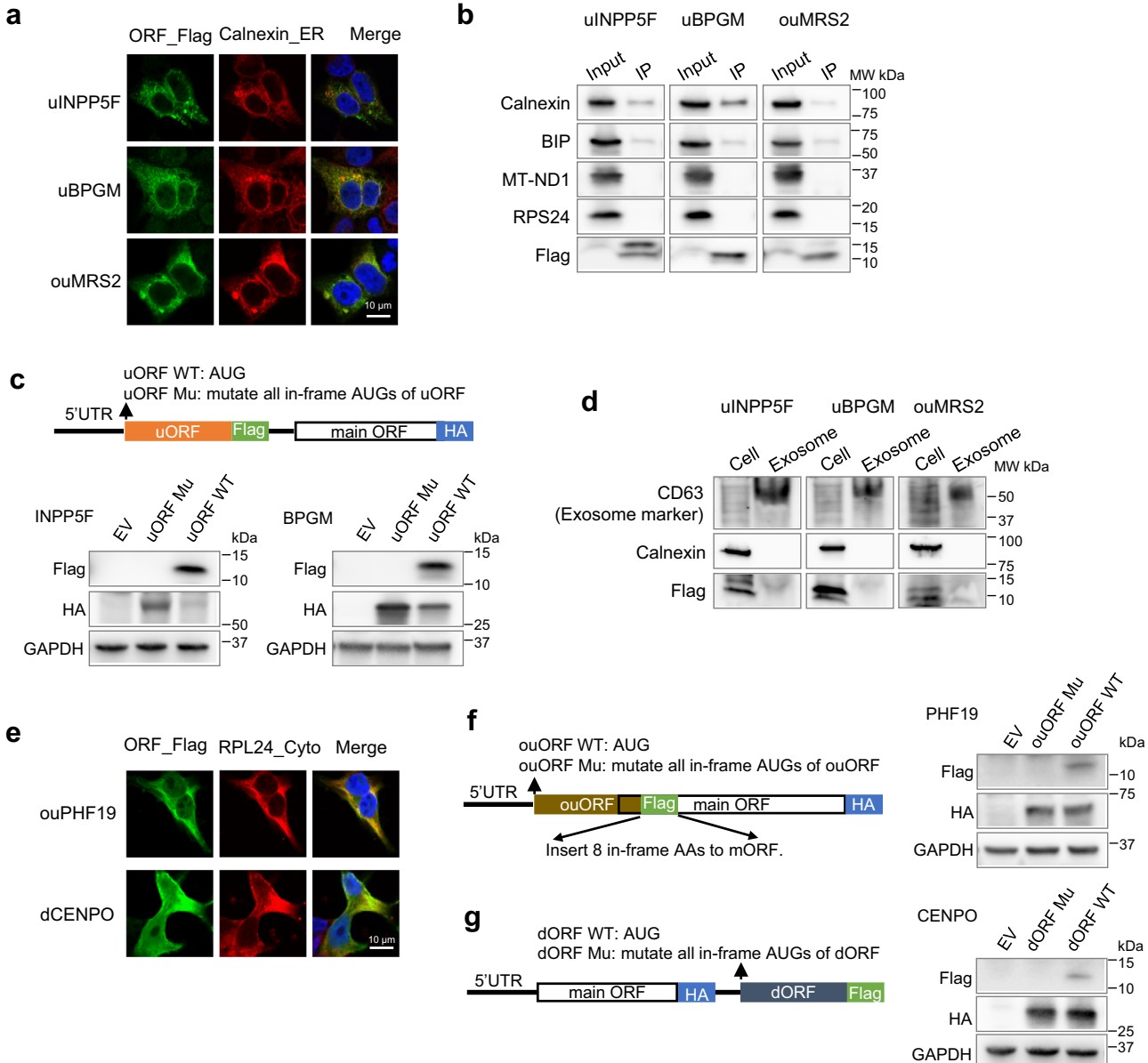

**Fig. 7 | Noncanonical peptides follow the conventional rules and localize to different subcellular compartments. a** The immunostaining experiments showing uINPP5F, uBPGM, and ouMRS2 (green) are localized to the ER. Calnexin (red) was used as the ER marker protein. The DAPI staining (blue) was used to label the cell nucleus. Scale bar, 10 μm. **b** Western blot analysis of uINPP5F, uBPGM, and ouMRS2 co-IP lysates. We examined the expression of ER proteins calnexin and BIP, mitochondrial protein MT-ND1, and cytosolic protein RPS24. Only ER proteins interact with the uORF peptides. **c** Western blot showing the expression of uINPP5F, uBPGM, and main ORFs in their native transcript. Flag-tagged uORF and HA-tagged main ORF were ectopically expressed in HEK293T cells. EV, empty vector control. **d** We performed western blot analysis of the exosome fraction and whole cell lysate to examine the expression of uINPP5F, uBPGM, and ouMRS2. CD63 protein was used as the exosome marker. The nonsecreted protein calnexin was used as the

whole-cell lysate maker. We detected uORF peptide expression in both exosome and whole cell lysates, indicating these peptides are secreted extracellularly. **e** Flag-tagged ORFs were ectopically expressed in HEK293T cells and co-immunostained for Flag (green), cytosolic marker RPL24 (red), and DAPI (blue). Scale bar, 10 μm. **f** Western blot showing the expression of ouPHF19 and PHF19 (main ORF) in the native transcript. Flag-tagged ouORF and HA-tagged main ORF were ectopically expressed in HEK293T cells. The flag tag induced an insertion of 8 in-frame amino acids into the main ORF without disturbing the protein sequences. **g** Western blot showing the expression of CENPO (main ORF) and dCENPO in the native transcript. dORF was tagged with a Flag and the main ORF was tagged with an HA. Experiments (**a–g**) were performed three times with similar results. Source data are provided as a Source Data file.

had PhyloCSF scores <0. Since poorly conserved lncRNAs can play various biological roles through functional RNA domains, these species-specific proteins can be biologically important. It will be interesting to dissect their detailed molecular functions.

PepScore is also correlated with peptide response to the inhibition of proteasomal and lysosomal protein degradation pathways. For unstable peptides with moderate PepScores, inhibiting these degradation pathways can drastically enhance peptide expression. Most

peptides with low PepScore (<0.3) did not show expression even after the combinatory inhibition of proteasome and lysosome pathways. One possibility is that these short peptides are directly degraded by proteases in the cell. The tens of protease types expressed in a cell make it difficult to block their activities simultaneously. A recent study showed that some noncanonical proteins can be degraded by the BAG6-mediated proteasome pathway[55]. Here we expressed the full-length translated ORFs in cells and showed that the peptides can be

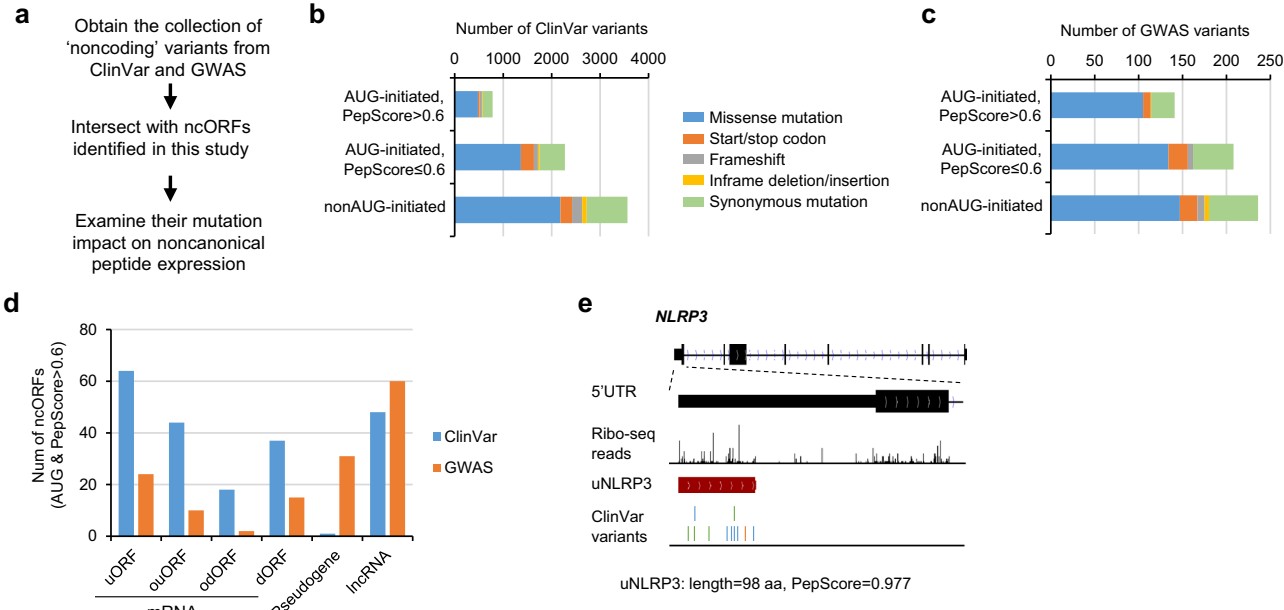

**Fig. 8 | Analyses of ClinVar and GWAS variants in ncORFs. a** Schematic illustrating the workflow for analyzing 'noncoding' variants annotated by ClinVar and GWAS catalog in ncORFs. **b** The number of ClinVar variants located in different ncORF types, grouped based on the start codon and PepScore. We used the software SnpEff to annotate different functional impacts of the variant on the ncORFs. **c** As in (**b**), the GWAS variants were analyzed. **d** For the ncORFs with AUG start codon and high PepScore (>0.6), we plotted the number containing ClinVar and GWAS variants grouped based on ORF type. **e** The example gene NLRP3 encodes a high-PepScore uORF containing ClinVar variants. The variants are colored based on mutation types shown in (**b**).

regulated by a variety of degradation pathways correlated with Pep-Scores (or lengths). While our PepScore prioritizes noncanonical peptides for experimental characterization, further studies are needed to examine the molecular properties of microproteins subject to different degradation pathways, such as motif existence or protein secondary structures. Learning these basic molecular mechanisms can add an additional layer to quantitatively predict cellular microprotein expression.

Because the majority of characterized functional proteins use AUG as their start codons and 5'-end methionine is critical for protein stabilization, our PepScore was trained only based on AUG-initiated ORFs. Our RibORF analyses also identified thousands of non-AUG-initiated ORFs that were translated in a cell. Currently, there lacks enough experimental data for these non-AUG ORFs to train a predictive model, although studies showed that a few CUG-initiated ORFs can have cellular functions. It was shown that both canonical Met-tRNA$_i$$^{Met}$ and Leu-tRNA$^{CUG}$ can decode non-AUG start codons[56–58]. However, for individual ORFs, their regulatory mechanisms remain elusive. Future experiments are needed to focus on non-AUG-initiated peptides and examine basic molecular mechanisms regulating their expression and stability.

Pervasive translation happens in annotated untranslated regions of mRNAs or off-frame regions (i.e., uORFs, dORFs, and iORFs). 1954 uORF/iORF/dORF peptides showed high PepScores (>0.6). Our experiments showed that many of them can be stably expressed in association with protein complexes. The ectopic expression of endogenous transcripts tagging both ncORFs and canonical ORFs showed that the two peptides can be generated confirming the polycistronic nature of the transcripts. These results raise the semantics issue of conventional gene structure definition based on the "one-gene one-polypeptide" hypothesis. Precautions are needed in the future characterization of individual candidate polycistronic RNAs, including checking the accuracy of transcript isoform annotations and performing experiments showing that the tandem peptides can be generated from one transcript isoform.

Our study provides a valuable resource for future functional studies of noncanonical peptides during development and disease. As high stability is an important feature for potential functions, the Pep-Score provides a quantitative guide to prioritize these peptides for detailed characterization. On the other hand, although most noncanonical peptides (>80%) are predicted to have low PepScores, they could function as regulators of main ORF expression (e.g., uORFs). The quickly degraded noncanonical peptides can also be presented by the MHCs and could be used as neoantigens for disease therapy[21–23]. Many ncORFs contain ClinVar and GWAS variants, which can be biologically and clinically important. Functional characterization of these noncanonical peptides in *human* diseases may lead to the development of novel therapeutic strategies.

## Methods

### Ribosome profiling data processing

We downloaded ribosome profiling datasets from the NIH Gene Expression Omnibus (GEO) and the EMBL-EBI European Nucleotide Archive (ENA) databases. The datasets we analyzed are listed in Supplementary Data 1. We trimmed sequencing adapters of reads and then mapped reads to ribosomal RNA (rRNA) sequences annotated in each respective species using Bowtie (v2.2.6)[59]. The non-rRNA reads were mapped to reference transcriptomes and genomes using TopHat (v2.1.0)[60]. We used GENCODE[61] transcriptome annotations for *human* (hg38, GENCODE V39) and *mouse* (mm10, GENCODE V25), and we used ENSEMBL gene annotation for *zebrafish* (GRCz11), *worm* (WBcel235), and *yeast* (sacCer3). The uniquely mappable reads were used for subsequent analyses. The samtools (v1.9)[62] and deepTools (v3.1.1)[63] were used for processing the sequencing reads.

### Identifying genome-wide translated ORFs using RibORF[30]

For each ribosome profiling sample, we grouped reads based on their fragment sizes and plotted the distribution of 5'-ends of fragments around start and stop codons of annotated mRNAs. We calculated the fraction of reads assigned to the 1st nucleotides of codons (in-frame

reads). We required that the high-quality reads should show in-frame reads >60%, and only these reads were used for downstream analyses. We calculated the offset distance to adjust the 5′-end locations of the reads to the ribosomal A-sites (the correction parameters are shown in Supplementary Data 1). These A-site adjusted reads were next used to examine genome-wide translated ORFs.

Based on transcript annotations in each genome, we identified candidate ORFs (with a start codon (NUG/AUC) and a stop codon) and required that ORFs included in the downstream analyses should contain >10 ribosome profiling reads. Suppose the ORF contained $L$ codons (excluding the start codon). For the codon $i$, the number of reads located in the 1st, 2nd, and 3rd nucleotides is $n_{i1}$, $n_{i2}$, and $n_{i3}$, respectively. We used the following three features to model whether the reads showed continuous 3-nt periodicity across codons in a candidate ORF: (1) the fraction of the reads assigned to 1st nt of codons:

$$f1 = \frac{\sum_{i=1}^{L} n_{i1}}{\sum_{i=1}^{L} n_{i1} + \sum_{i=1}^{L} n_{i2} + \sum_{i=1}^{L} n_{i3}}$$; (2) the fraction of codons supporting in-frame translation (i.e., the 1st nucleotide contains more reads than the 2nd and 3rd): $f2 = \sum_{i=1}^{L} (n_{i1} > n_{i2} \,\&\, n_{i1} > n_{i3})/L$; and (3) the PME value to measure the uniformness of read distribution across the codons as detailed below[13,64].

For each ORF with $L$ codons, supposed the total read number is $N$. We divide the ORF into smaller regions based on $N$ and $L$: if $N > L$, we define a region length as 1 codon; otherwise, a region length is defined as floor $(L/N)$. For each region $j$ in an ORF, we calculated the fraction of reads in the region: $P(X_j) = n_j/N$, where $n_j$ represents the number of reads in region $j$. We then calculated the entropy value of the read distribution: $E = \sum_{j=1}^{m} (P(X_j) * \log_2 P(X_j))$, where $m$ is the total number of regions. The PME value is calculated as: $f3 = E/maxE$, where $maxE$ is the entropy value when we resampled the reads to be perfectly evenly distributed across codons in an ORF.

We trained a logistic regression model to identify translated ORFs using the three features ($f1$, $f2$, and $f3$). A candidate ORF has a higher probability of being actively translated if the three features show higher values. We used canonical ORFs as positive examples. Internal off-frame ORFs and those showing highly localized distribution with PME ($f3$) < 0.1 (likely representing non-ribosomal RNA-protein complexes[64]) were used as negative examples. We randomly selected 1000 positive examples and 3000 negative examples to train the model with 5-fold cross-validation. To test the performance of the classifier, we used another 1000 positive and 3000 negative examples for the testing dataset and calculated the AUROC.

To account for lineage-specific regulation, we built a logistic regression classifier to identify translated ORFs for each cell type from each ribosome profiling dataset (Supplementary Data 2). We also built a model using merged reads from each species. To minimize the false positive predictions, we required that an ORF should show positive predictions ($P > 0.6$) by two independent models. We further used Salmon (v1.6.0)[31] to calculate the ORF isoform level expression and required that an ORF should show expression level RPKM (reads per kilobase of exon per million reads mapped) >0.2 in at least one cell type.

## Comparing the ORF predictions using other published software

To examine the robustness of our RibORF predictions, we used several other published software to predict genome-wide translated ORFs in *human*, including RiboTish[32], RiboCode[33], PRICE[34], and RiboTricer[35]. We performed a prediction for each high-qualify ribosome profiling sample showing strong 3-nt periodicity in Supplementary Data 1. RiboTish, RiboCode, and RiboTricer allowed the customized parameters for the offset correction distance between the 5′-end of reads and ribosomal P-sites. The distances were based on the parameters listed in Supplementary Data 2: the offset distance to P-site = the

distance to A-sites − 3. We performed a prediction using an AUG-start codon only and another prediction allowing NUG/AUC start codons. The minimal ORF length was set to 2 codons. For PRICE, it does not allow customized parameters, the prediction was done using the default parameters.

Then we compared our RibORF prediction with those obtained from these software. We annotated ORFs with perfect ORF structure match. Additionally, as these algorithms used different approaches to select representative start codons of an ORF, we also annotated ORFs showing stop codon match and start codon mismatch across the predictions. These comparison results were shown in Supplementary Data 5.

## The collection of microproteins

To examine the ORF features of stable microproteins, we curated a list of studied peptides from the literature/database. We selected microproteins <100 aa and with an AUG start codon, and required ORF sequences show a perfect match between our annotation vs. published. We collected 343 stable microproteins from the following three sources: (1) RefSeq-defined microproteins, which have well-characterized functional roles. The longest protein isoform of a gene was considered. (2) Stable microproteins based on the ectopic expression experiments from Prensner et al. [18]. In the study, they fused the full-length noncanonical ORFs with a V5-tag and the same 5′UTR/3′UTR, and performed the ectopic expression in HEK293T cells. They performed in-cell immunoblotting to quantify the peptide expression. If the expression value >1, a peptide was considered to be stably expressed. (3) Complex-associating microproteins from Chen et al. [19]. They used the co-IP mass spectrometry experiments and showed that the peptides are associated with other proteins. There are overlaps of these microproteins from three sources, shown in the Venn diagram (Supplementary Fig. 5a).

We collected 100 microproteins which were undetectable based on ectopic expression experiments from Prensner et al. [18] with expression values <1. The list of stable and undetectable peptides is shown in two different spreadsheets in Supplementary Data 6.

In this list, we only considered microproteins encoded by putative lncRNAs or canonical ORFs of mRNAs. Some of the peptides studied in Prensner et al. [18] or Chen et al. [19] were not included if they were longer than 100 aa, used non-AUG start codon, had sequence mismatches with our annotation, or were annotated as ncORFs (e.g., uORFs) in mRNAs.

## Annotation of molecular features of translated ORFs

We used PhyloCSF scores based on the genome alignment across 58 mammals to measure the peptide conservation level. DeepLoc[43] was used to predict the subcellular localization of the peptides. We calculated the amino acid composition of a peptide sequence using categories of hydrophobic ("F", "L", "I", "V", "P", "A", "G"), charged ("R", "K", "D", "E"), polar ("Q", "N", "H", "S", "T", "C"), hydrophilic ("C", "D", "E", "H", "K", "N", "Q", "R"), and amphipathic ("W", "Y", "M"). The following codons showing high usage frequency compared to other synonymous ones were considered optimized codons: "GCC", "CGC", "AAC", "GAC", "TGC", "CAG", "GAG", "GGC", "CAC", "CTG", "AAG", "CCC", "TTC", "AGC", "ACC", "TAC", and "GTG". We used the tool pyHCA to perform the hydrophobic cluster analysis (HCA) and examine peptide foldability[39].

We used Pfam (v1.6-2)[40] and TMHMM (v2.0c)[41] to search whether a protein domain exists in a peptide sequence. We calculated the motif strength surrounding AUG start codons (6 nt upstream of the AUG and 1 nt downstream of the AUG) based on the consensus Kozak sequence. The Kozak score is calculated as described in Chen et al. [19]: if the −6 position is a 'G', +3; if the −5 position is a 'C', +1; if the −4 position is a 'C', +1; if the −3 position is a 'G' or 'A', +3; if the −2 position is a 'C', +1; if the −1 position is a 'C', +1; and if the +1 position is a 'G', +3. We took a 40 nt

sequence around start codons (20 nt upstream and 20 nt downstream) and used ViennaRNA (v2.5.0)[65] to fold the RNA sequence and calculate the minimum free energy (MFE) value.

To study the human noncanonical peptide conservation in other species, we used BLASTP (v2.6.0)[66] to examine the sequence similarities in ORFs we identified other species. We consider a human peptide to be conserved if it meets the following criteria: (1) the two ORFs have a BLASTP alignment E-value < $10^{-4}$; (2) the ORF types are the same; (3) the start codons are the same.

We used the Tau-index[42] to calculate the tissue-specific expression across the following 11 cell/tissue types: B-LCL: GSE143263, Liver: E-MTAB-7247, HepG2: GSE129061, Brain: E-MTAB-7247, Testis: E-MTAB-7247, iPSC: GSE131650, K562: GSE129061, A375: GSE143263, HCT116: GSE143263, HeLa: GSE125218, and HEK293T: GSE125218. Suppose $x_i$ is the expression level of the gene in tissue $i$, and $n$ is the number of samples. We calculated the relative expression of the gene as $r_i = x_i / \max_{1 \leq i \leq n}(x_i)$. The Tau index is calculated as $\tau = \frac{\sum_{i=1}^{n}(1-r_i)}{n-1}$.

### Calculating the length FDRs of translated ORFs

To obtain the expected ORF lengths in a transcript, we generated 1000 pseudo-transcripts of the same length from shuffled transcript sequences and random genome sequences. For each pseudo-transcript, we identified all possible ORF structures with an AUG start codon and a stop codon. Then we calculated the length distribution of these expected ORFs from pseudo-transcripts. Suppose there are $n$ total possible ORFs from a pseudo-transcript; for each of them, we examined whether its length is longer than the observed translated ORF in the transcript. Suppose $m$ ORFs giving rise to a pseudo-transcript are longer than the observed length. The FDR of the observed ORF length was calculated as $m/n$. The FDR values calculated based on the shuffled transcripts and random genome sequences were significantly correlated (Supplementary Fig. 5f). We then used the merged expected ORFs from the two sets of randomized sequences for the final FDR calculation.

### Building the logistic regression model PepScore to predict peptide stable probability

Based on the comparisons of stable vs. undetectable microproteins, we found genomic features showing a significant difference between the two types. Three ORF features were included in our PepScore model: (1) the FDR of ORF length: $t1 = -log_{10}(FDR\ value)$; (2) the existence of a Pfam or transmembrane (TMHMM) domain ($t2 = 1$ if there is a domain; $t2 = 0$ if no identified domain); (3) conservation measured by the PhyloCSF score as $t3$. Suppose an ORF has $K$ codons; for each codons $i$ the in-frame PhyloCSF score is $f_i$. $t3$ is calculated as $\log 10(|\sum_{i=1}^{K} f_i| + 1) * sign(\sum_{i=1}^{K} f_i)$. In our final model, we used log(sum of PhyloCSF scores) across the ORF region to model the conservation. We also tried using the average PhyloCSF score, but found it performed worse than the sum value.

Based on these features ($t1$, $t2$, and $t3$), we trained a logistic regression model, PepScore, to calculate each peptide's stable probability, using the R (v3.5.1) package "caret". We randomly selected 200 microproteins for the training dataset with 5-fold cross-validation, and a different set of 100 microproteins for the testing dataset. The R command line used for 5-fold cross-validation is "train_control ← trainControl(method = "cv", number = 5)", and the command for building the logistic regression model is "model ← train(as.factor(stability) ~ t1 + t2 + t3, data = input, trControl = train_control, method = "glm", family=binomial())". For our final model, the training and testing sets show comparable AUROC values in distinguishing stable vs. undetectable microproteins. We used the command "evalmod" of the R package "precrec" to calculate the AUROC values. Adding more ORF features to the classifier, as presented in Supplementary Fig. 6, did not increase the prediction accuracy (the AUROC change was <0.01). To compare PepScore's performance in classifying the two types of microproteins vs. using other ORF features, we calculated AUROC using the individual ORF features to perform the classification.

### Cell culture

HEK293T (CRL 3216) and MCF-7 (HTB-22) cells were purchased from the American Type Culture Collection. Cells were cultured in Dulbecco's modified Eagle medium (DMEM) supplemented with 10% fetal bovine serum (FBS) and maintained in a humidified incubator at 37 °C with 5% $CO_2$.

### ORF ectopic expression and expression level quantification

To generate plasmids used for ectopic expression, the ORFs were PCR amplified, digested, and then inserted into the pcDNA3.1(+) vector (Invitrogen) with a fused Flag tag before the stop codon. The CDS of EGFP was cloned into the pcDNA3.1(+) vector and was used as a positive control. All CDSs of ORFs and primers used for amplification were synthesized by Twist Bioscience and Integrated DNA Technologies (IDT). We listed ORF sequences and primers in Supplementary Data 8.

To examine the expression of ncORFs in their native transcript context, we cloned the full transcript (5'UTR to the stop codon of main ORF for uORFs, ouORFs, and iORFs; 5'UTR to the stop codon of dORF for dORFs) to the pCDH-CMV-MCS-EF1-Puro vector (System Biosciences) through the Nhe1 and Not1 restriction sites by gene assembly. NcORFs were fused with a Flag tag and main ORFs were fused with an HA tag. Control plasmids were generated by mutating all in-frame start codons in ncORFs. All full transcript sequences and primers are listed in Supplementary Data 8.

To ectopically express ORFs, $1.5 \times 10^5$ HEK293T cells were grown on poly-L-lysine (Sigma) coated 15 mm cover glasses placed in a 24-well plate for 12 h and transfected with 0.6 µg plasmid using lipofectamine 2000 reagent. Media was refreshed 6 h after transfection, and cells were further cultured for 24 h. Cells were fixed with 4% paraformaldehyde for 10 min at room temperature and washed twice with Dulbecco's phosphate-buffered saline (DPBS). Cells were then permeabilized with 0.1% Triton X-100 for 10 min and washed twice with DPBS. After blocking with 2.5% BSA for 1 h at room temperature, cells were stained with anti-Flag antibody (1:500, P1804, Sigma) overnight at 4 °C. Cells were washed three times and then incubated with secondary antibody (1:1000, A32723, Goat anti-Mouse Alexa Fluor 488, Invitrogen) for 1 h at room temperature. Cells were washed and mounted onto glass slides using Prolong™ Glass antifade mountant with NucBlue™ Stain (P36983, Invitrogen). We used 0.6 µg pcDNA3.1-EGFP plasmid as the positive control and 0.6 µg pcDNA3.1 empty vector as the negative control. Control samples were prepared in parallel. Images were captured using a Nikon A1R confocal microscope.

To quantify the expression level of ORFs, we analyzed and calculated the percentage of cells with a positive-Flag-staining signal by ImageJ. The percentage of GFP-positive cells was calculated, and the expression score of each ORF was given by normalizing the ORF-positive percentage to the GFP-positive percentage. More than 6000 cells from at least four experimental replicates were analyzed for each ORF.

To examine the magnitude of overexpression, total RNA was isolated after 24 h ectopic expression using the Direct-zol RNA Mini-Prep kit (Zymo Research). Reverse-transcript was carried out using a High-Capacity cDNA Reverse Transcription Kit (4368814, Applied Biosystems). Quantitative PCR was performed with PowerUp™ SYBR™ Green Master Mix (A25742, Applied Biosystems) on QuantStudio 3 (Applied Biosystems). All qPCR primers are listed in Supplementary Data 8.

## Protein degradation pathway inhibition

To inhibit peptide degradation, 6 h after transfection, cell culture media was changed to complete DMEM supplemented with proteasome inhibitors MG132 (M7449, Sigma; 5 μM) and lactacystin (2267, TOCRIS; 20 μM), and/or lysosome/autophagy inhibitors chloroquine (C-237, Sigma; 50 μM) and bafilomycin A1(S1413, Selleckchem; 50 nM).

## Generation of stable cell lines

To generate plasmids used for stable expression of ORFs in the various cell lines, we cloned the CDSs of the indicated ORFs (uSLC35A4, uMKKS, and iPGRMC1) into the pCDH-EF1-MCS-IRES-copGFP vector (System Biosciences) through the Nhe1 and Not1 restriction sites with a fused Flag tag before the stop codon. Control plasmids were generated by mutating all in-frame start codons. All CDSs and primers are listed in Supplementary Data 8.

To generate stable cell lines, lentivirus was produced by transfecting HEK293T cells with pMD2.G, pPAX2, and ORF-containing pCDH-EF1-ORF-IRES-copGFP plasmids. Media containing lentivirus was collected after 48 h and filtered through a 0.45 μm PVDF membrane. MCF-7 cells were spinfected in virus-containing media supplemented with 8 μg/mL polybrene at 1000 × g for 1 h at 37 °C. Media was replaced after overnight incubation. MCF-7 cells stably expressing ORFs were sorted by gating GFP-expressing cells on a BD Influx cell sorter after 3–4 days. Cells were expanded, and the ORF expression was confirmed by western blotting and immunostaining (all antibodies used are listed in Supplementary Data 8).

## CRISPR knockout

To generate plasmids used for CRISPR knockout, sgRNAs targeting uSLC35A4/PMPCA/PMPCB and nontargeting controls were cloned into the LentiCRISPRv2 backbone. sgRNAs are listed in Supplementary Data 8. All plasmids used in this study were verified through Sanger sequencing.

To generate knockout cells, lentivirus particles were packaged by transfecting HEK293T with pMD2.G, pPAX2, and constructed pLenti-CRISPRv2 plasmids. Media was changed after 6 h, and media containing lentivirus was collected after two days. The collected media was centrifuged at 2000 rpm for 5 min and filtered through a 0.45 μm PVDF membrane. MCF-7 cells expressing the indicated ORFs were spinfected in virus-containing media supplemented with 8 μg/mL polybrene at 1000 × g for 1 h at 37 °C. After 24 h, cells were selected with 1 μg/mL puromycin for 5 days. Protein knockout was validated with western blotting (antibodies are listed in Supplementary Data 8).

## Identification of peptide subcellular localization by immunostaining

The immunofluorescence was performed as described above, except for the antibody incubation. After blocking, cells were stained with anti-Flag *mouse* antibody (F1804, Sigma, 1:200) and co-stained with organelle markers for mitochondria, ER, or cytoplasm overnight (ab186735, anti-TMM20, 1:200, Abcam; 5318, Cell Signaling Technology, anti-AIF, 1:200; 12175-1-AP, anti-OXCT1 Proteintech, 1:100; 2679 T, anti-Calnexin, Cell Signaling Technology, 1:200; 17082-1-AP, anti-RPL24, Proteintech, 1:100). In Figs. 5b, 6c, Supplementary Fig. 9f and 9k, MCF-7 cells with stable ORF expression were incubated in secondary antibody solution: Alexa Fluor 555 Goat anti-Mouse (A21424, Invitrogen, 1:1000) and Alexa Fluor 647 Goat anti-Rabbit (A32733, Invitrogen, 1:1000). In Figs. 7a, 7e, Supplementary Fig. 9a, 9b, 10a, 10b, 10f, 10h, and 10i, ORF-expressing HEK293T cells were incubated in a secondary antibody solution: Alexa Fluor 488 Goat anti-Mouse (A32723, Invitrogen, 1:1000) and Alexa Fluor 555 Goat anti-Rabbit (A21428, Invitrogen, 1:1000) for 1 h at room temperature. After washing, cells were mounted with Prolong™ Glass antifade mountant with NucBlue™ Stain. Slides were imaged on a Nikon A1R confocal microscope.

## Identification of peptide interaction partners by immunoprecipitation/mass spectrometry

HEK293T cells were seeded on a 6-cm dish at 60% confluency and transfected with 4 μg ORF-containing pcDNA3.1 plasmids the next day. Cell media was changed after 6 h and cells were collected after 48 h. Peptides were immunoprecipitated with anti-Flag M2 magnetic beads (M8823, Sigma) according to the manufacturer's instructions. Briefly, washed cells were lysed in lysis buffer (50 mM Tris HCl, pH 7.4, 150 mM NaCl, 1 mM EDTA, 1% Triton X-100, and protease inhibitor) for 10 min on ice. After clarification, the supernatant was transferred to a new centrifuge tube and incubated with pre-washed anti-Flag M2 magnetic beads overnight in an agitator at 4 °C. Beads were washed three times with buffer (50 mM Tris HCl, pH 7.4, 150 mM NaCl), and peptides were eluted with SDS-PAGE sample buffer. Samples were boiled and loaded into a 4–20% Mini-protein TGX stain-free gel (Bio-Rad). The stacking gel was prepared by running the gel at 120 V for 5 min until all samples were embedded into the gel. The gel was stained with Coomassie blue and the protein-containing region was excised. The peptide/protein complexes were then subjected to in-gel digestion and prepared for mass spectrometry. We used an empty vector as the negative control, and each peptide was prepared and measured in triplicate. The enrichment of peptides was validated with western blotting.

Raw mass spectrometry files were processed with MaxQuant (1.6.17.0)[67] using label-free quantification and searched against a custom FASTA file. An FDR of 1% was applied to peptide-spectrum match (PSM) and protein level. The fixed modification was carbamidomethyl (C). Selected variable modifications were oxidation (M) and acetyl (protein N-terminus). Exported data were further analyzed with Perseus (v1.6.15.0)[68]. After data normalization, filtering, and imputation, interactors were identified by performing two-sample two-sided t-tests. The enrichment values and *P*-values were exported for plotting.

## Analysis of published mass spectrometry and CRISPR screening data

We downloaded published mass spectrometry datasets (PXD019486, PXD020620, PXD014031; 188 samples in total) from nine *human* cell lines (e.g., iPSCs, HEK293T, PC3), six *mouse* cell lines (e.g., mESCs, MEF, 4T1), and eight *mouse* tissues (e.g., brain, kidney, heart)[19,69]. The raw data were processed with MaxQuant (v1.6.17.0) using label-free quantification and searched against a customized FASTA file with the ORF sequences identified in this study. For whole proteome data, peptides with a minimum length of eight amino acids were considered for the search, including N-terminal acetylation and methionine oxidation as variable modifications and cysteine carbamidomethylation as a fixed modification. Enzyme specificity was set to trypsin-specific. A FDR of 1% was applied to PSM and protein levels. A maximum of two missed cleavages was allowed. Maximum precursor and fragment ion mass tolerance were set to 4.5 and 20 ppm, respectively.

For the MHC I-bound peptides, we downloaded datasets from six allotype-resolved cell lines[70] and three DLBCL cell lines[21] (PXD000394, PXD020620; 31 samples in total). When searching HLA and MCH peptidome data, an FDR of 1% was applied to PSM. Protease specificity was set to nonspecific, and possible peptide identifications were restricted to 8-15 amino acids. We also incorporated the MHC I-detected peptides shown in Ouspenskaia et al.[22]. To examine whether an ORF knockout impacts cell growth, we analyzed the phenotypic scores presented in Chen et al.[19], who designed a customized CRISPR library to knock out lncRNA ORFs and uORFs and examined their impact on iPSCs and K562 cells. A lower phenotypic score indicated stronger inhibition of cell proliferation.

## Mitochondrial membrane potential measurement

MCF-7 cells and MCF-7_uSLC35A4 cells with uSLC35A4 knockout or control knockout were seeded in 8-well μSlides (80826, iBidi) at 35,000 cells/well. When cells were grown to 80% confluency the next

day, CCCP (positive control of mitochondrial membrane potential loss; 13296, Cell Signaling Technology) was added to the control wells to a 50 μM final concentration. Cells were incubated at 37 °C for 30 min. To measure mitochondrial membrane potential, TRME solution (13296, Cell Signaling Technology) was added to each well to a final concentration of 100 nM, and cells were stained in an incubator for 20 min. Cells were washed three times with warmed PBS and then incubated in pre-warmed FluoroBrite DMEM (A18967, Gibco). Live images were acquired on a Nikon A1R confocal microscope. Parameters of image capture were set based on CCCP-treated and untreated MCF-7 cells.

For analysis of the membrane potential of mitochondria in cells in different experimental conditions, the intensities of TMRE (tetramethylrhodamine, ethyl ester) fluorescence were measured with ImageJ software (v1.52) following published methods[71,72]. Statistical analyses were performed by unpaired two-sided Student's $t$-tests using the Prism analysis program (GraphPad v9.2.0).

### Exosome isolation

Extracellular vesicles were isolated using Total Exosome Isolation Reagent (4478359, Invitrogen) following the manufacturer's instructions. Briefly, HEK293T cells were grown on a 10-cm dish to 80% confluency and transfected with 10 μg pcDNA3.1 plasmids. After 6 h, culture media was changed to DMEM supplemented with 10% exosome-depleted FBS (A2720803, Giboco) and protease inhibitors (P1860, Sigma). Cell culture media was collected after two days and centrifuged at 2000 × $g$ for 30 min. The supernatant was transferred to a new tube and mixed with 0.5 volumes of the total exosome isolation reagent. Samples were incubated at 4 °C overnight and centrifuged at 10,000 × $g$ for 1 h at 4 °C. The pellet was suspended in 1x Laemmli sample buffer and analyzed by western blotting. CD63 (10628D, Invitrogen) was used as an exosome marker[73].

### Western blotting

Cells were lysed in RIPA buffer (89900, Thermo) supplemented with a protease inhibitor cocktail (Roche). Protein concentrations were determined by a Detergent Compatible Bradford Assay kit (23246, Thermo). Calibrated samples were diluted with 4x Laemmli sample buffer (Bio-Rad), and equal amounts of total protein were separated in Mini-protein TGX stain-free gels (Bio-Rad). Proteins were transferred to nitrocellulose membranes using a Trans-blot Turbo Transfer System (Bio-Rad). The membranes were blocked with 5% nonfat milk in TBST, incubated with primary antibodies overnight at 4 °C, washed three times with TBST at room temperature, incubated with HRP secondary antibodies, and imaged using the Bio-Rad Chemidoc imaging system. All antibodies and dilution information used in this study are listed in Supplementary Data 8.

### RNA library preparation

RNA-seq libraries were generated as the previously described[74]. In brief, total RNA was extracted with a Direct-zol RNA kit (Zymo Research), and mRNAs were then isolated using Oligo(dT)25 magnetic beads (New England Biolabs) according to the manufacturer's instructions. Purified mRNAs were fragmented with a NEBNext Mg$^{2+}$ RNA fragmentation module (New England Biolabs) at 94 °C for 4 min. Fragmented RNAs were then precipitated overnight at −20 °C by adding 0.1 volume of 3 M sodium acetate, 10 mg of glycoblue, and 1.2 volumes of isopropanol. RNA fragments were re-suspended, and the library was constructed with A-tailing and SMARTer oligo-based template switching method.

### RNA-seq data analyses

We trimmed the 3′ sequencing adapters from the reads (AAAAAAAA for the A-tailing methods). For the libraries we used SMARTer oligo-based template switching, and we trimmed the first 7 nt, including the random 4 nt and the 3 locked Gs in the 5′ sequencing adapters. The trimmed reads were mapped to the reference genome (hg38) and the GENCODE-defined transcriptome using STAR (v2.1.0)[75]. We used HTSeq (v0.9.1)[76] to generate gene-level read counts. Genes showing dynamic regulation of RNA expression after ectopic ORF expression were defined based on the following criteria: (1) transcript per million (TPM) >3 in at least one condition; (2) >1.3-fold expression change in both replicates.

### Gene ontology analyses

Gene ontology analyses were conducted using the DAVID database[77].

### Genetic variant analyses

We downloaded genetic variant annotations from the ClinVar database[50] and the NHGRI-EBI GWAS calatog[49]. We filtered out these located in RefSeq-defined coding regions. Next, we used the software SnpEff (v4.5)[78] to intersect and annotate the variants in *human* ncORFs we identified in this study.

### Reporting summary

Further information on research design is available in the Nature Portfolio Reporting Summary linked to this article.

## Data availability

We analyzed a large-cohort of published ribosome profiling datasets and their accession numbers are listed in Supplementary Data 1. The sequencing datasets generated in this study are available in the Gene Expression Omnibus (GEO) repository with the accession number GSE216093. The mass spectrometry data were deposited in the Proteomics Identifications Database (PRIDE) with the project accession PXD037658. Source data are provided with this paper.

## Code availability

Computational codes were deposited in GitHub (https://github.com/zhejilab/RibORF and https://github.com/zhejilab/PepScore).

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

## Acknowledgements

This work was supported by grants to Z.J.: the National Institutes of Health (R35GM138192, R01HL161389, and R00CA207865), and the Lynn Sage Scholar fund. E.S. was supported by the Predoctoral Training Program in Biomedical Data Driven Discovery (T32LM012203). Proteomics services were performed by the Northwestern Proteomics Core Facility, supported by NCI CCSG P30 CA060553 awarded to the Robert H Lurie Comprehensive Cancer Center, instrumentation award (S10OD025194) from NIH Office of Director, and the National Resource for Translational and Developmental Proteomics supported by P41 GM108569. We thank the members of the Ji lab for helpful discussions.

## Author contributions

H.Y., Q.L. and Z.J. conceived and designed the study. H.Y., Q.L., E.S., S.W. and Z.J. performed the data analysis. Q.L. performed experiments. Q.L. and Z.J. wrote and revised the manuscript. Z.J. supervised the research.

## Competing interests

The authors declare no competing interests.
