## [Peer Review File · Nature Communications]

Widespread subspecies-specific stable noncanonical peptides identified by integrated analyses of ribosome profiling and ORF featuresREVIEWER COMMENTS

Reviewer #1 (Remarks to the Author):

The manuscript by Yang, Li et al "Integrated analysis of ribosome profiling and ORF features identify stable noncanonical peptides in humans" is a resubmission of a manuscript originally submitted to Nature Methods. The manuscript is much improved, experimentally, but I still have minor concerns. Though, I don't think it's appropriate to request additional experiments at this stage of the revision. My main concern, which is shared with the other reviewer, is still with regard to the significance of the paper and whether it merits publication in Nature Communication. But I can be persuaded either way, so I will leave it to the editor and other reviewer(s) to decide.

The paper's main "advance" is the development of PepScore, but the algorithm represents no conceptual advance. "ORF length", "with a protein domain", and "PhyloCSF score" are so obvious features. The authors claim that their use of "logistic regression" is novel, but it basically just combines the 3 (and other) features into a single "score" (and logistic regression is basically just one line in R). And, if ORF length FDR, ORF length, protein domain, and PhyloCSF already are at >0.8 AUROC, do future researchers really need to calculate PepScore? Again, I want to emphasize that PepScore is still a useful tool, since comparing a single score is better than comparing 3 or 4 different numbers, but I am just not sure whether it's significant enough of an advance. I will leave it to the other reviewer(s) to decide.

Following from my point above, the authors claim that PepScore "identifies stable noncanonical peptides", and "effectively predicts native peptide expression and subsequent regulation via protein degradation pathways". I can see how ORF length, conservation, presence of protein domain can be "correlated" with protein stability, but I don't see the logic of how these input features can "directly predict" something that requires amino acid properties and secondary structure properties. PMID: 37046090 is a good recent study actually showing regulation via protein degradation pathways. Maybe the authors should reword some of their claims, especially since the authors' overall writing style tends to over-claim their findings, which can mislead future readers (an issue also pointed out by the other reviewer).

Similarly, for the section on ClinVar and GWAS, are AUG-initiated PepScore>0.6 ORFs with ClinVar/GWAS significantly enriched? Figure 7 only shows the numbers, but the % should be shown instead. If not significantly enriched, what is the purpose of the entire section on pg 18 and 19?

Finally, I raised this issue in my previous review. The EDARADD dORF does not look real. I checked GWIPs-viz, and instead of a uniform coverage of ribosome reads, there's regions with a lot of reads and regions with no reads at all. This looks more like an artifact from multimapping reads (or other reasons) rather than a real ORF. Either the authors demonstrate that the dORF can be detected via Western (which the author doesn't show), or the authors remove the section on that dORF.

Reviewer #2 (Remarks to the Author):

In this manuscript, Yang and colleagues describe a comprehensive analysis of public ribosome profiling data from multiple eukaryotes in search of novel short Open Reading Frames (sORFs). The authors generated a catalog of translated sequences from five species, identified features that correlated with previously reported ORFs whose peptides could be detected, and used these features to classify their predicted sORFs as "stable" and "unstable". Contrary to previous studies, they report that "stable" ORFs are poorly conserved. In addition, the authors expressed some predicted ORFs exogenously and found that they could be detected and localized subcellularly using immunostaining. Finally, they describe how many of these sORFs have ClinVar variants. This manuscript is timely in that it falls into the broad goal of annotating genomic sequences to find novel proteins, which is a topic of current

research in many groups. The cell line experiments convincingly show that some sORFs can be expressed. However, the logistic regression modeling approach, PepScore, appears to have a major flaw that creates circular reasoning, as described below.

1) The authors present Pepscore as a logistic regression model to determine whether peptides are "stable" or "unstable". To do this, they start with lists of "stable" and "unstable" sORF proteins from two previous studies, Prensner et al., and Chen et al.. The logistic regression classifier then built a regression classifier to separate the two classes. The main feature separating the classes is ORF length, which seems like it is likely artifactual given the initial way that "stable" and "unstable" peptides were classified by the published studies.

The Prensner and Chen papers didn't truly assay protein stability (i.e. half-life). Rather, they primarily determined whether peptides from proteins were detected by mass-spectrometry after trypsin digestion. Mass-spectrometry has a natural bias towards longer ORFs. Longer ORFs are more likely to have trypsin digestion sites (Lysine or Arginine, but not Arg-Pro and Lys-Pro). Longer ORFs are also more likely to produce more fragments of protein than shorter ORFs. Because of this, longer ORFs are generally more likely to be detectable by mass-spectrometry than shorter ORFs. An analogy would be to use RNA-seq. If you used an enzyme to fragment RNA only after G residues (e.g. RNase T1) and sequenced the fragments, longer RNAs and more G-rich RNAs would be more likely to have a fragment that was "detectable". This is essentially the case with mass-spectrometry. Indeed, the Chen et al paper notes this, "MS-based proteomics using tryptic digestion identified far fewer noncanonical peptides, which may be due to challenges in detecting the trypsin digested products from short, noncanonical CDSs or possibly to more rapid turnover of these noncanonical peptides". So detectable sORFs tend to be longer probably because they generate more peptide fragments. There is no evidence that this is due to protein stability, half-life, etc. The regression classifier just picks up on the feature that correlates with MS-detection probability (again, not stability). the FDR of ORF length value most likely correlates directly with ORF length. This is a major flaw in the study, as it suggests Pepscore is artifactually classifying sORFs as "stable" and "unstable" based on their length simply because mass-spec is biased towards detecting longer ORFs.

2) There are also well-documented biases in Mass-spec sample preparation that are likely to be exaggerated with shorter proteins. Overall, classifying proteins as "stable" or "unstable" based on detection by mass-spec is probably not very accurate.

3) The sORFs used to train the classifier are not clearly described in the manuscript, or in table S6. The manuscript says that 70 "stable" peptides were discovered in Prensner et al, and 100 "unstable" peptides were found in the Prensner and Chen papers. Prensner et al reported mass-spectrometry detection of 174 of 553 tested sORFs. They also detected 257 / 553 sORFs by overexpression and microscopy. How did the authors get to a list of 70, and which ones are those? Similarly, which "unstable" sORFs were taken from each of the two studies? How were they defined by each study? Later, the authors say that there were 343 stable and 100 unstable microproteins, but it's not clear where those were reported. Also, the manuscript methods section is inconsistent with the results, in saying that there were 300 total microproteins (200 for trainin and 100 for testing). All of these peptides are supposedly listed in table S6, but that table has 443 total sORFs listed (which matches the 343 + 100, but not the 70 + 100). This is very disorganized and very confusing, and precludes an independent replication of the results. The authors need to clearly state which proteins were reported as "stable" and "unstable" in previous work and cite the study in which that classification was found.

4) There are a number of places where the manuscript exaggerates the novelty of the results. Typically, this is where the authors use the word "reveal", which should only be used if something truly unknown to the field was shown for the first time. For example, in the abstract, the authors state that ectopic expression "revealed stable complex-associating microproteins can be encoded in 5'/3' untranslated regions and overlapping coding regions of mRNAs besides annotated noncoding RNAs". This was arguably "revealed" by the Chen et al paper. The first section of the results is titled

"Ribosome profiling data analyses revealed the complexity of mammalian translome". This is hype. Earlier ribosome profiling studies (circa 2010-2013) did this. We already know the mammalian translome is complex and includes lncRNAs, sORFs and pseudogenes. Indeed the section ends with, "These results underscored the complexity of mammalian translome", not "reveal". The title would be more accurate to simply say "A comprehensive catalog of ORFs from five eukaryotic species". Similarly, "Our study revealed pervasive translation in annotated untranslated regions of mRNAs or off-frame regions (i.e., uORFs, dORFs, and iORFs)." This was already known. It's not revealed in this particular study.

Point-by-point Response

Reviewer #1 (Remarks to the Author):

The manuscript by Yang, Li et al “Integrated analysis of ribosome profiling and ORF features identify stable noncanonical peptides in humans” is a resubmission of a manuscript originally submitted to Nature Methods. The manuscript is much improved, experimentally, but I still have minor concerns. Though, I don’t think it’s appropriate to request additional experiments at this stage of the revision. My main concern, which is shared with the other reviewer, is still with regard to the significance of the paper and whether it merits publication in Nature Communication. But I can be persuaded either way, so I will leave it to the editor and other reviewer(s) to decide.

We thank the reviewer for thinking that our manuscript was much improved compared to the last version. Following we addressed the reviewers’ further concerns.

The paper’s main “advance” is the development of PepScore, but the algorithm represents no conceptual advance. “ORF length”, “with a protein domain”, and “PhyloCSF score” are so obvious features. The authors claim that their use of “logistic regression” is novel, but it basically just combines the 3 (and other) features into a single “score” (and logistic regression is basically just one line in R). And, if ORF length FDR, ORF length, protein domain, and PhyloCSF already are at >0.8 AUROC, do future researchers really need to calculate PepScore? Again, I want to emphasize that PepScore is still a useful tool, since comparing a single score is better than comparing 3 or 4 different numbers, but I am just not sure whether it’s significant enough of an advance. I will leave it to the other reviewer(s) to decide.

Our PepScore is the first one to integrate the three features and generate a probabilistic value to predict that an ORF encodes a stable peptide. We analyzed different ORF features and narrowed them down to these three specific ones to build the logistic regression. Before we generated the model, it was unclear how well the combination of ORF features or individual ones could distinguish stable vs. unstable noncanonical peptides. The Δ AUROC is >0.07 comparing PepScore vs. individual features, which is not trivial in the genomic analyses. I would argue that a simplified model/protocol using only the relevant features is better than complicated uninterpretable models. Much diligence and detailed thought were into building this seemed easy model.

Importantly, our model showed the relative contribution of three ORF features. The ORF length FDR showed a more significant contribution to the prediction, compared to the PhyloCSF score and with a protein domain. We performed systematic validation and showed many lowly conserved ORFs from 5’/3’ UTRs and overlapping coding regions of mRNAs can encode stable complex-associating peptides. This represents an unexpected novel biological finding because high conservation has been used as a major criterion to identify candidate functional proteins.

Our study integrated the comprehensive identification of translated ORFs across five eukaryotic species, the development of PepScore, as well as systematic follow-up molecular experiments. I believe the whole piece of our work represents a major advance in understanding how stable noncanonical peptides are encoded in genomes and provide a valuable resource to the scientific community.

Following from my point above, the authors claim that PepScore “identifies stable noncanonical peptides”, and “effectively predicts native peptide expression and subsequent regulation via protein degradation pathways”. I can see how ORF length, conservation, presence of protein domain can be “correlated” with protein stability, but I don’t see the logic of how these input features can “directly predict” something that requires amino acid properties and secondary structure properties. PMID: 37046090 is a good recent study actually showing regulation via protein degradation pathways. Maybe the authors should reword some of their claims, especially since the authors’ overall writing style tends to over-claim their findings, which can mislead future readers (an issue also pointed out by the other reviewer).

We thank the reviewer for the detailed suggestions. We now rephrased the sentences:

1. Changed the title to “Identifying genomic stable noncanonical peptides by integrated analyses of ribosome profiling and ORF features”.
3. “Our systematic follow-up experiments validated the correlation between PepScore and peptide stability.”

Following I would like to explain the differences between our study and Kesner et al (PMID: 37046090). In Kesner et al, the major experimental system they used was fusing a proportion of noncanonical ORF sequences (30 aa or 13 aa, and they did not require the ORFs to be actively translated by analyzing ribosome profiling data) to the C-terminal end of the EGFP ORF (239 aa). They found the role of BAG6 in inducing the proteasome degradation of these peptides. They further showed that the pathway regulates the stability of AMD1 readthrough protein.

In our study, we expressed the full-length translated ORF sequences in cells with different PepScores/lengths, which is different from Kesner et al. We found the correlation between PepScore and peptide stability. Some noncanonical peptides are stable. For unstable ones, the ones with moderate PepScores can be stabilized upon the inhibition of either proteasomal or lysosomal pathways. But for the very short ones with low PepScores, the combination inhibition of proteasomal and lysosomal pathways cannot rescue their cellular expression suggesting that those peptides can be directly degraded by proteases. The Kesner et al paper does not diminish the novelty of our work but emphasizes that our work is timely performing experiments using the endogenous full-length noncanonical ORFs. We showed that some noncanonical peptides are stable and diverse protein degradation pathways can regulate noncanonical peptide stability/degradation. We now cited the Kesner et al paper and elaborated on this in the Discussion section.

Similarly, for the section on ClinVar and GWAS, are AUG-initiated PepScore>0.6 ORFs with ClinVar/GWAS significantly enriched? Figure 7 only shows the numbers, but the % should be shown instead. If not significantly enriched, what is the purpose of the entire section on pg 18 and 19?

We now added the fraction of the ncORF groups with ClinVar or GWAS variants. The AUG-initiated PepScore>0.6 ORFs show a comparable fraction containing ClinVar variants vs. other ncORF groups, but a higher fraction containing GWAS variants (Fig. S11a-b).

The major reason we presented these genetics analyses in the manuscript is that ClinVar and GWAS variants are not randomly occurring ones. Many ClinVar variants are associated with human diseases with uncertain significance/biological reason. The GWAS variants are significantly associated with

human tracts based on the genome sequencing analyses. However, for many GWAS variants, their impacts on gene expression remain to be determined. Our analyses pointed out that some disease/trait-associated variants can regulate noncanonical translated ORFs. Our purpose was to raise this question and provide a resource/catalog to the community.

Finally, I raised this issue in my previous review. The EDARADD dORF does not look real. I checked GWIPs-viz, and instead of a uniform coverage of ribosome reads, there's regions with a lot of reads and regions with no reads at all. This looks more like an artifact from multimapping reads (or other reasons) rather than a real ORF. Either the authors demonstrate that the dORF can be detected via Western (which the author doesn't show), or the authors remove the section on that dORF.

As suggested, we now deleted the section describing the EDARADD dORF.

Reviewer #2 (Remarks to the Author):

In this manuscript, Yang and colleagues describe a comprehensive analysis of public ribosome profiling data from multiple eukaryotes in search of novel short Open Reading Frames (sORFs). The authors generated a catalog of translated sequences from five species, identified features that correlated with previously reported ORFs whose peptides could be detected, and used these features to classify their predicted sORFs as "stable" and "unstable". Contrary to previous studies, they report that "stable" ORFs are poorly conserved. In addition, the authors expressed some predicted ORFs exogenously and found that they could be detected and localized subcellularly using immunostaining. Finally, they describe how many of these sORFs have ClinVar variants. This manuscript is timely in that it falls into the broad goal of annotating genomic sequences to find novel proteins, which is a topic of current research in many groups. The cell line experiments convincingly show that some sORFs can be expressed. However, the logistic regression modeling approach, PepScore, appears to have a major flaw that creates circular reasoning, as described below.

We thank the reviewer for appreciating the significance of our work. Following we addressed all reviewers' questions.

1) *The authors present Pepscore as a logistic regression model to determine whether peptides are "stable" or "unstable". To do this, they start with lists of "stable" and "unstable" sORF proteins from two previous studies, Prensner et al., and Chen et al.. The logistic regression classifier then built a regression classifier to separate the two classes. The main feature separating the classes is ORF length, which seems like it is likely artifactual given the initial way that "stable" and "unstable" peptides were classified by the published studies.*

The Prensner and Chen papers didn't truly assay protein stability (i.e. half-life). Rather, they primarily determined whether peptides from proteins were detected by mass-spectrometry after trypsin digestion. Mass-spectrometry has a natural bias towards longer ORFs. Longer ORFs are more likely to have trypsin digestion sites (Lysine or Arginine, but not Arg-Pro and Lys-Pro). Longer ORFs are also more likely to produce more fragments of protein than shorter ORFs. Because of this, longer ORFs are generally more likely to be detectable by mass-spectrometry than shorter ORFs. An analogy would be to use RNA-seq. If you used an enzyme to fragment RNA only after G residues (e.g. RNase T1) and sequenced the fragments, longer RNAs and more G-rich RNAs would be more likely to have a fragment that was "detectable". This is essentially the case with mass-spectrometry. Indeed, the Chen

et al paper notes this, "MS-based proteomics using tryptic digestion identified far fewer noncanonical peptides, which may be due to challenges in detecting the trypsin digested products from short, noncanonical CDSs or possibly to more rapid turnover of these noncanonical peptides". So detectable sORFs tend to be longer probably because they generate more peptide fragments. There is no evidence that this is due to protein stability, half-life, etc. The regression classifier just picks up on the feature that correlates with MS-detection probability (again, not stability). the FDR of ORF length value most likely correlates directly with ORF length. This is a major flaw in the study, as it suggests Pepscore is artifactually classifying sORFs as "stable" and "unstable" based on their length simply because mass-spec is biased towards detecting longer ORFs.

Our grouping of stable vs. unstable microproteins did not rely on the detection by mass spectrometry. We are sorry about the confusion and realized our original description in the manuscript lacked details.

We collected 343 stable microproteins (<100 aa and with an AUG-start codon) from the following three sources:

- (1) 273 RefSeq-defined microproteins, which have well-characterized functional roles.
- (2) 77 stable microproteins based on the ectopic expression experiments from Prensner et al. In the study, they fused the full-length noncanonical ORFs with a V5-tag and performed the ectopic expression in HEK293T cells. They performed in-cell immunoblotting to quantify the peptide expression. If the expression value >1, a peptide was considered to be stable. Their experimental design is similar to the follow-up experiments we performed in this study except that we used the Flag-tag.
- (3) 3 complex-associating microproteins from Chen et al in which the co-IP mass spectrometry experiments showed that the peptides are associated with other proteins.

There are overlaps of these microproteins from three sources, as shown in the following Venn diagram.

We collected 100 unstable microproteins which were based on ectopic expression experiments from Prensner et al with expression values <1.

We now added the detailed description in the Methods section and the Venn diagram to Fig. S5a.

2) *There are also well-documented biases in Mass-spec sample preparation that are likely to be exaggerated with shorter proteins. Overall, classifying proteins as "stable" or "unstable" based on detection by mass-spec is probably not very accurate.*

See the above answer to the question 1. Our definition of “stable” or “unstable” was not based on the detection by mass-spec.

3) *The sORFs used to train the classifier are not clearly described in the manuscript, or in table S6. The manuscript says that 70 "stable" peptides were discovered in Prensner et al, and 100 "unstable" peptides were found in the Prensner and Chen papers. Prensner et al reported mass-spectrometry detection of 174 of 553 tested sORFs. They also detected 257 / 553 sORFs by overexpression and microscopy. How did the authors get to a list of 70, and which ones are those? Similarly, which "unstable" sORFs were taken from each of the two studies? How were they defined by each study? Later, the authors say that there were 343 stable and 100 unstable microproteins, but it's not clear where those were reported. Also, the manuscript methods section is inconsistent with the results, in saying that there were 300 total microproteins (200 for trainin and 100 for testing). All of these peptides are supposedly listed in table S6, but that table has 443 total sORFs listed (which matches the 343 + 100, but not the 70 + 100). This is very disorganized and very confusing, and precludes an independent replication of the results. The authors need to clearly state which proteins were reported as "stable" and "unstable" in previous work and cite the study in which that classification was found.*

We thank the reviewer for the detailed comment. Indeed, our original writing and table were confusing. We only selected a subset of noncanonical ORFs from Prensner and Chen Papers based on the following criteria. First, we required the peptide length to be shorter than 100 aa. We focused on analyzing these microproteins in this study. Second, we required the ORFs to have an AUG start codon. Third, we required a perfect sequence match (100%) of the ORF sequences between our annotation and theirs. Fourth, we required the ORFs to be from annotated lncRNAs or be annotated canonical ORFs in RefSeq-defined coding genes (i.e. the major ORF of a transcript).

Some ORFs from Prensner et al and Chen et al were not included, if they were longer than 100 aa, used a non-AUG start codon, had sequence mismatches with our annotation, or were annotated as noncanonical ORFs (e.g. uORFs) from mRNAs.

To train the logistic regression model, we did not include all stable and unstable peptides. We randomly selected 200 peptides for training and a separate 100 peptides for testing. As described in the last rebuttal letter, the model is quite robust if we repeat the analyses by randomly sampling or using the same number of stable and unstable peptides (Person Correlation coefficient of predicted PepScore values >0.98).

To address this, we now modified Table S6, split the microproteins into two separate spreadsheets (i.e. one for stable peptides and another for unstable ones), and added appropriate column annotations. We also modified the methods section to describe the above criteria and showed the numbers in a Venn diagram (Fig. S5a).

4) *There are a number of places where the manuscript exaggerates the novelty of the results. Typically, this is where the authors use the word "reveal", which should only be used if something truly unknown to the field was shown for the first time. For example, in the abstract, the authors state that ectopic expression "revealed stable complex-associating microproteins can be encoded in 5'/3' untranslated regions and overlapping coding regions of mRNAs besides annotated noncoding RNAs". This was arguably "revealed" by the Chen et al paper. The first section of the results is titled "Ribosome profiling data analyses revealed the complexity of mammalian translome". This is hype. Earlier ribosome*

profiling studies (circa 2010-2013) did this. We already know the mammalian translome is complex and includes lncRNAs, sORFs and pseudogenes. Indeed the section ends with, "These results underscored the complexity of mammalian translome", not "reveal". The title would be more accurate to simply say "A comprehensive catalog of ORFs from five eukaryotic species". Similarly, "Our study revealed pervasive translation in annotated untranslated regions of mRNAs or off-frame regions (i.e., uORFs, dORFs, and iORFs)." This was already known. It's not revealed in this particular study.

We did not mean to claim we are the first when we wrote: "reveal". As the reviewer suggested, we now rephrased the sentences/words.

Finally, we would like to thank both reviewers for the detailed comments and critical reading of our manuscript, which significantly improved our work.

REVIEWERS' COMMENTS

Reviewer #1 (Remarks to the Author):

Overall, the manuscript is good work. But, I don't think my concerns and the other reviewer's concerns regarding the significance of the manuscript has been fully addressed. As I stated in my previous review, I will leave the final decision to the editor.

I disagree with the author's argument that "a simplified model/protocol using only the relevant features is better than complicated uninterpretable models. Much diligence and detailed thought were into building this seemed easy model." Similar types of models have been built in previous publications (though just not for predicting peptide stability). Furthermore, it is not like the authors started with hundreds of possible features and only the "obvious" ones were significant at the end; the authors started from very obvious features such as "ORF length", "Kozak", "conservation", "codon optimization", etc to begin with. Lastly, the training data is from a very small sample (343), and is likely from a biased selection (curated examples from previous publications), so I am not sure how generalizable and useful the model will be for future researchers.

Practically speaking, the model does not provide new biology regarding the ORFs. And, I am also skeptical how useful the model will be for future researchers. ORF length, conservation, and protein domain are such strong filters and features that all researchers will look at. And, the presence of any feature will be a good hint that the ORF will be a good candidate for follow up experiments, so I am not sure how much added benefit the model will bring (considering the caveats of the model from the limited training data).

Lastly, the new title doesn't make much sense. What is a "genomic stable noncanonical peptides"? What is a "genomic peptide"?

Reviewer #2 (Remarks to the Author):

I appreciate the extra details provided by the authors for this manuscript. Unfortunately, the clarifications about the training data used for the PepScore model raise additional concerns, primarily that the model classifies proteins as being either more like "annotated Ref-seq short ORFs" or more like "unexpressable" short ORFs from Prensner et al. This is not a classification of "stable" vs "unstable" or "functional" vs "non-functional" as the authors have framed the model. This is elaborated below:

1) Rather than training the PepScore model on "stable" and "unstable" microproteins, the authors trained the model to distinguish between Ref-seq annotated proteins less than 100 amino acids (273) combined with sORFs that were ectopically expressed in Prensner et al (77) vs sORFs that Prensner et al did not see by ectopic expression (100). The positive set is overwhelmingly comprised of annotated short ORFs (79.5%) while the negative set are exclusively from ectopic expression studies. It seems this comparison was originally defined as a way to model "functional" vs "non-functional" short proteins (from the first submission). The authors changed the wording to "stable" and "unstable" in response to reviewer concerns. I have strong concerns about using Ref-seq short ORFs as most of the "positive" set, as ORF length and other features may be artificially higher in this annotated set of short proteins from Ref-seq than in the <100 amino acid subset of ORFs detected by ribosome profiling from Prensner. The resulting statistical model separates annotated from inexpressible, not necessarily stable vs unstable. This reduces the apparent novelty of the work because the statistical model implications are more limited than the manuscript claims.

2) It seems the authors should redo their modeling using only the data from Prensner et al, including the 257 "stable" and 296 "unstable" ectopic proteins. Assuming that the 177 they used earlier are simply the ORFs less than 100 amino acids, they could also redo the modeling using only those. Such modeling would provide apples-to-apples comparisons at least and provide a model that predicts the ability to detect ectopically expressed noncanonical ORFs.

3) If using the Prensner data alone does not allow the authors to build a functional classifier, they should at least revise the manuscript to actually reflect what the model does - identify characteristics that distinguish annotated Ref-seq sORFs from inexpressible, unannotated sORFs from Prensner. Pepscore classifies RibORF predictions based on whether they are more similar to annotated sORFs or to inexpressible sORFs; this is not stable vs unstable per se. This needs to be very clearly explained to readers, along with a discussion of the caveats associated with the model. Perhaps it would be fair to say that Pepscore helps prioritize sORFs most likely to be expressed, but it certainly doesn't necessarily reflect stability or functionality. This would require a lot of revision throughout the manuscript to clarify the actual meaning of Pepscore probabilities.

Point-by-point response

Reviewer #1 (Remarks to the Author):

Overall, the manuscript is good work. But, I don't think my concerns and the other reviewer's concerns regarding the significance of the manuscript has been fully addressed. As I stated in my previous review, I will leave the final decision to the editor.

I disagree with the author's argument that "a simplified model/protocol using only the relevant features is better than complicated uninterpretable models. Much diligence and detailed thought were into building this seemed easy model." Similar types of models have been built in previous publications (though just not for predicting peptide stability). Furthermore, it is not like the authors started with hundreds of possible features and only the "obvious" ones were significant at the end; the authors started from very obvious features such as "ORF length", "Kozak", "conservation", "codon optimization", etc to begin with. Lastly, the training data is from a very small sample (343), and is likely from a biased selection (curated examples from previous publications), so I am not sure how generalizable and useful the model will be for future researchers.

We now performed more analyses showing the robustness of our PepScore, which was suggested by Reviewer 2. We divided the microproteins into two subgroups based on their sources: the RefSeq-defined ones and those stably expressed from ectopic experiments (Prensner et al). First, we showed that the ORF features we selected show consistent patterns comparing the two subgroups vs. the negative set (undetectable from ectopic expression), including the longer ORF lengths, higher conservation and with an annotated domain. Second, we built a new logistic regression model classifying the detectable vs. undetectable ectopically expressed peptides alone. The generated *P*-values from the new model are well correlated with our PepScore with a Pearson correlation coefficient value of 0.98. And we further showed the advantage of our PepScore, which can identify short, conserved ones with a domain. These analyses were detailed in the response to Reviewer 2's comments, and further demonstrated the robustness of our model. Importantly, we performed systematic validation experiments using independent uORF peptides and showed the correlation between PepScore and peptide stability.

We acknowledged that other protein features can regulate peptide stability. We discussed that "While our PepScore prioritizes noncanonical peptides for experimental characterization, further studies are needed to examine the molecular properties of microproteins subject to different degradation pathways, such as motif existence or protein secondary structures. Learning these basic molecular mechanisms can add an additional layer to quantitatively predict cellular microprotein expression."

Practically speaking, the model does not provide new biology regarding the ORFs. And, I am also skeptical how useful the model will be for future researchers. ORF length, conservation, and protein domain are such strong filters and features that all researchers will look at. And, the presence of any feature will be a good hint that the ORF will be a good candidate for follow up experiments, so I am not sure how much added benefit the model will bring (considering the caveats of the model from the limited training data).

As I described in the last response, our PepScore is the first to convert the ORF features into a probabilistic value. Although the features we used are not surprising as the reviewer pointed out, it was unclear which cutoffs should be used based on individual ORF features or feature combinations to identify candidate noncanonical ORFs encoding stable peptides. I believe our PepScore provides unique guidance for prioritizing the noncanonical ORFs/peptides for further experimental characterization.

Lastly, the new title doesn't make much sense. What is a "genomic stable noncanonical peptides"?
What is a "genomic peptide"?

We now changed our title to "Genome-wide identification of stable noncanonical peptides by integrated analyses of ribosome profiling and ORF features".

Reviewer #2 (Remarks to the Author):

I appreciate the extra details provided by the authors for this manuscript. Unfortunately, the clarifications about the training data used for the PepScore model raise additional concerns, primarily that the model classifies proteins as being either more like "annotated Ref-seq short ORFs" or more like "unexpressable" short ORFs from Prensner et al. This is not a classification of "stable" vs "unstable" or "functional" vs "non-functional" as the authors have framed the model. This is elaborated below:

1) Rather than training the PepScore model on "stable" and "unstable" microproteins, the authors trained the model to distinguish between Ref-seq annotated proteins less than 100 amino acids (273) combined with sORFs that were ectopically expressed in Prensner et al (77) vs sORFs that Prensner et al did not see by ectopic expression (100). The positive set is overwhelmingly comprised of annotated short ORFs (79.5%) while the negative set are exclusively from ectopic expression studies. It seems this comparison was originally defined as a way to model "functional" vs "non-functional" short proteins (from the first submission). The authors changed the wording to "stable" and "unstable" in response to reviewer concerns. I have strong concerns about using Ref-seq short ORFs as most of the "positive" set, as ORF length and other features may be artificially higher in this annotated set of short proteins from Ref-seq than in the <100 amino acid subset of ORFs detected by ribosome profiling from Prensner. The resulting statistical model separates annotated from inexpressible, not necessarily stable vs unstable. This reduces the apparent novelty of the work because the statistical model implications are more limited than the manuscript claims.

2) It seems the authors should redo their modeling using only the data from Prensner et al, including the 257 "stable" and 296 "unstable" ectopic proteins. Assuming that the 177 they used earlier are simply the ORFs less than 100 amino acids, they could also redo the modeling using only those. Such modeling would provide apples-to-apples comparisons at least and provide a model that predicts the ability to detect ectopically expressed noncanonical ORFs.

3) If using the Prensner data alone does not allow the authors to build a functional classifier, they should at least revise the manuscript to actually reflect what the model does - identify characteristics that distinguish annotated Ref-seq sORFs from inexpressible, unannotated sORFs from Prensner. Pepscore classifies RibORF predictions based on whether they are more similar to annotated sORFs or to inexpressible sORFs; this is not stable vs unstable per se. This needs to be very clearly explained to readers, along with a discussion of the caveats associated with the model. Perhaps it would be fair to say that Pepscore helps prioritize sORFs most likely to be expressed, but it certainly doesn't necessarily reflect stability or functionality. This would require a lot of revision throughout the manuscript to clarify the actual meaning of Pepscore probabilities.

The grouping of microproteins

Here the reviewer raised a new question regarding our microprotein grouping and comparisons.

I think that our collection of stable microproteins from the two major sources is valid. (1) The expressed peptides from the Prensner study should be stable, as quickly degraded ones cannot be detected by the ectopic expression; (2) The RefSeq-defined microproteins are stable as well. They have well-

defined functions and regulate specific biological pathways. Scientists produced antibodies and/or used the tagged proteins to show they are expressed when characterizing their functional roles.

This will not affect us asking the question of which ORF features distinguish stable microproteins vs. those undetectable by ectopic expression. And I still agree with Reviewer 1's comment from the first round that the most direct microprotein feature we examined in this study is "stability" but not "function", although the two features are correlated and high stability is crucial for potential peptide function.

We now changed our negative set name from "unstable" to "undetected by the ectopic expression". We discussed that these undetectable peptides are possibly "unstable" because their ORF translation was driven by the same vector with the same 5'/3' UTRs vs. detectable peptides.

The performance of PepScore

Another reviewer's concern is whether our PepScore is biased toward the classification of our negative set ("undetected by ectopic expression") vs. positive subgroups, as there are more RefSeq-defined microproteins than those detected by ectopic expression from the Prensner study. In the updated Supplementary Figure 6, we plotted the distribution of peptide lengths, phyloCSF scores (measuring the peptide conservation across mammals), and the fraction with an annotated domain among the groups. Compared to the negative set, the microproteins defined by RefSeq or detectable by ectopic expression all show longer lengths, higher phyloCSF scores, and a higher fraction with annotated domains (Supplementary Figure 6a, 6l, and 6m). The trend is consistent between two positive subgroups, although RefSeq-defined ones have a more significant trend. The two positive sets all show significantly higher PepScores than the negative set ($P < 10^{-10}$ comparing detectable vs. undetectable from ectopic expression and AUROC=0.79 classifying the two groups; $P < 10^{-46}$ comparing RefSeq-defined vs. undetectable and AUROC=0.99 classifying the two groups) (Supplementary Figure 6n, 6o, and 6p). The data showed that two stable microprotein subgroups show common characteristics of our selected ORF features and our PepScore can classify the negative set vs. both positive sets.

Additionally, we performed the analyses as the reviewer suggested using the Prensner dataset alone to build a logistic regression model (with 5-fold cross-validation) to classify the detectable (77) vs. undetectable microproteins (100) from ectopic expression. As shown in the Response Figure 1a, the ORF length FDR still made the most significant contribution to the prediction, while conservation levels especially with a domain showed less contribution to the prediction. This is because the Prensner set analyzed much fewer conserved peptides and/or those with an annotated. We examined the correlation between the predicted probabilities from the new model vs. the original PepScores and obtained a quite good correlation coefficient of 0.98 (Response Figure 1b). Most peptides showed comparable predicted values between the two models. Using the predicted probabilities from the new model, the AUROC is 0.79 classifying detectable vs. undetectable microproteins from ectopic expression which is the same as using our original PepScores (Response Figure 1c). And the AUROC is 0.98 classifying RefSeq-defined microprotein vs. the undetected group, which is slightly lower than that from our original PepScore (Response Figure 1d). A small number of microproteins showed low predicted probabilities in the new model (<0.6) but high original PepScores (>0.6) (Response Figure 1b). This is because these short peptides have high PhyloCSF scores (conservation) and with an annotated domain. These include well-characterized functional stable microproteins such as (25 aa), SLN (31 aa), and APELA (54 aa) (Response Figure 1b).

The data indicated that adding the RefSeq-defined microproteins to the training set allowed our PepScore to identify the conserved ones with an annotated domain. As I detailed in the above response, we have legitimate reasons to compile the two sources of peptides as the stable group. We keep our original PepScore.

Altogether, I think the reviewer's new concern does not change our conclusion that PepScore identifies the "stable" microproteins. Importantly, we performed systematic follow-up experiments and validated the correlation between peptide stability and PepScore.

Response Figure 1. Build a logistic regression model using detectable vs. undetectable ectopically expressed microproteins only. (a) The model parameters. (b) The correlation between the predicted probabilities between the new model vs. our original PepScores for the collected microproteins. (c) AUROC showing the performance using the new model to classify the detected vs. undetected peptides from ectopic expression. (d) AUROC showing the performance using the new model to classify RefSeq-defined proteins vs. undetected peptides from ectopic expression.